# The Occurrence of Mycotoxins in Raw Materials and Fish Feeds in Europe and the Potential Effects of Deoxynivalenol (DON) on the Health and Growth of Farmed Fish Species—A Review

**DOI:** 10.3390/toxins13060403

**Published:** 2021-06-05

**Authors:** Paraskevi Koletsi, Johan W. Schrama, Elisabeth A. M. Graat, Geert F. Wiegertjes, Philip Lyons, Constanze Pietsch

**Affiliations:** 1Aquaculture and Fisheries Group, Wageningen University and Research, 6708 WD Wageningen, The Netherlands; johan.schrama@wur.nl (J.W.S.); geert.wiegertjes@wur.nl (G.F.W.); 2Adaptation Physiology Group, Wageningen University and Research, 6708 WD Wageningen, The Netherlands; lisette.graat@wur.nl; 3Alltech Biotechnology Inc., A86 X006 Dunboyne, Ireland; plyons@alltech.com; 4School of Agricultural, Forest and Food Sciences (HAFL), Applied University Berne (BFH), 3052 Zollikofen, Switzerland

**Keywords:** mycotoxins, survey, wheat, maize (corn), soybean meal, fish feed, deoxynivalenol (DON), fish, growth, toxic effects

## Abstract

The first part of this study evaluates the occurrence of mycotoxin patterns in feedstuffs and fish feeds. Results were extrapolated from a large data pool derived from wheat (*n* = 857), corn (*n* = 725), soybean meal (*n* = 139) and fish feed (*n* = 44) samples in European countries and based on sample analyses by liquid chromatography/tandem mass spectrometry (LC-MS/MS) in the period between 2012–2019. Deoxynivalenol (DON) was readily present in corn (in 47% of the samples) > wheat (41%) > soybean meal (11%), and in aquafeeds (48%). Co-occurrence of mycotoxins was frequently observed in feedstuffs and aquafeed samples. For example, in corn, multi-mycotoxin occurrence was investigated by Spearman’s correlations and odd ratios, and both showed co-occurrence of DON with its acetylated forms (3-AcDON, 15-AcDON) as well as with zearalenone (ZEN). The second part of this study summarizes the existing knowledge on the effects of DON on farmed fish species and evaluates the risk of DON exposure in fish, based on data from in vivo studies. A meta-analytical approach aimed to estimate to which extent DON affects feed intake and growth performance in fish. Corn was identified as the ingredient with the highest risk of contamination with DON and its acetylated forms, which often cannot be detected by commonly used rapid detection methods in feed mills. Periodical state-of-the-art mycotoxin analyses are essential to detect the full spectrum of mycotoxins in fish feeds aimed to prevent detrimental effects on farmed fish and subsequent economic losses for fish farmers. Because levels below the stated regulatory limits can reduce feed intake and growth performance, our results show that the risk of DON contamination is underestimated in the aquaculture industry.

## 1. Introduction

Aquaculture, in contrast to capture fisheries that have remained stable over the last decades, continues to grow and contribute to the increasing food supply for human consumption, reaching worldwide production of 80 million metric tonnes (Mt) in 2016 [1]. To sustain its growth, the aquaculture industry is highly dependent on commercial feed sources [2,3,4]. Indeed, the production of aquafeeds increased from 8 Mt in 1995 to 48 Mt in 2015 [1]. A recent global feed survey revealed that the annual growth of aquafeed production for 2018 was 4% [5], and was projected to reach 65 Mt in 2020 [6]. However, the inclusion rate of traditionally used finite and expensive marine protein and fat sources from wild-caught fish (i.e., fishmeal and fish oil) in the diets of farmed fish species will continue to decline and the industry has already shifted to crop-based ingredients to meet the rising demand for aquafeeds [2,6,7]. For instance, collective data from the Norwegian salmon (*Salmo salar*) industry reflect the change in modern aquaculture diet composition and confirm the reduced dependency on fishmeal derived from wild-caught fish; while in 1990 salmon diets consisted of 90% marine ingredients, already in 2013 their inclusion rate was less than 30%, which increased the share of plant protein sources to 37% [8]. Plant-based ingredients increasingly replace marine-based ingredients and, therefore, an enhanced level of understanding of the nutritional quality of raw materials derived from plant sources is becoming increasingly important for aquafeeds.

Plant-based feed ingredients currently used in aquafeeds as substitutes for marine ingredients include soybean meal, rapeseed/canola meal, maize/corn, wheat bran and wheat [3]. Even in diets for carnivorous species like Atlantic salmon, the main protein and lipid sources used within the feed in 2012 were derived from crops, such as soybean meal (21.3% average inclusion rate) and rapeseed oil (18.3% average inclusion rate), with the main starch source being wheat (9.9% average inclusion rate) [8]. However, in contrast to marine ingredients that contain well-balanced protein contents to meet the amino acid requirements of aquatic farmed animals, the continuing transition towards higher inclusion of plant-based ingredients poses a real challenge for aquafeed producers due to nutritional limitations [9,10]. The higher inclusion of less-expensive plant sources may introduce a series of anti-nutritional factors (e.g., protease inhibitors, phytates, saponins, glucosinolates, tannins, non-starch polysaccharides) and/or increase the occurrence of animal feed contaminants; factors that might affect the quality and safety of aquafeeds [11,12,13,14,15]. Frequently occurring natural feed contaminants are mycotoxins, which are mainly detected in plant-based feedstuffs [16,17,18,19,20]. Increasingly [21,22,23,24,25], the presence of mycotoxins is reported in aquafeeds.

### 1.1. Mycotoxin-Producing Fungi

Mycotoxins are secondary metabolites produced by fungi that invade crops in the field during plant growth and/or fungi that colonize the crops before harvest and predispose the commodity to mycotoxins after harvest during drying, transportation and storage [19,26]. Common toxigenic genera are *Aspergillus*, *Penicillium*, *Fusarium, Alternaria* and *Claviceps* which proliferate with climatic conditions considered favourable (close to their preferred temperature and moisture) [27,28,29]. The global distribution of mycotoxigenic fungi is temperature-dependent; *Penicillium* spp. are common in cool climates, *Aspergillus* spp. in the tropics and *Fusarium* spp. in temperate areas [30]. Fungal growth requirements for minimal and optimal water activity (a_w_) differ among genera. *Fusarium* and *Alternaria* are plant pathogens and hygrophilic (1.00 a_w_), meaning they proliferate in substrates with high water availability and, therefore, predominate in the fields at pre-harvest. *Aspergillus* and *Penicillium* are xerophilic (<0.95 a_w_), meaning they can proliferate at low water availability and are the main mycotoxigenic fungi post-harvest, during storage [31]. Post-harvest measures such as proper storage conditions can possibly prohibit the growth of xerophilic fungi [32] but pre-harvest conditions such as a continuously changing climate [33] cannot be controlled, for which reason the presence and growth of hygrophilic fungi from the fields remains unpredictable.

The occurrence of mycotoxigenic fungi, however, does not necessarily lead to the production of mycotoxins. For instance, *Aspergillus* spp. were detected in aquafeed samples but not the corresponding mycotoxins [34]. Such observations reinforce questions of “How, why and when do fungi produce mycotoxins?” These respective questions largely remain unanswered since most research is focused on the toxicological aspects of mycotoxins and their effects on host organisms [35]. Mycotoxin production may be triggered after environmental abiotic stimuli (light, nutrient, pH) and biotic interactions of different microbes (i.e., fungal–bacterial or fungal–fungal) that lead to up-regulation of biosynthetic gene clusters to secure the ecological niche of fungi in hostile environments by exhibiting antimicrobial functions [36]. Indeed, incubation of commercial fish feeds under different storage conditions can influence fungal growth and mycotoxin production. Specifically, the application of warm (temperature ~ 27 °C) and humid (~70% relative humidity) conditions may trigger the release of ochratoxin A (OTA), with variations due to distinct hotspots with optimal conditions for fungal growth and production of mycotoxins [37]. Therefore, the presence of mycotoxigenic fungi under storage conditions does not necessarily mean the presence of mycotoxins in aquafeeds.

### 1.2. Classification of Fusarium Mycotoxins: “Traditional”, “Emerging” and “Masked”

*Fusarium* species as soil-borne microbes are the most common pathogens in cereal crops flourishing in a wide geographic range, also in Europe [38,39]. The toxicologically most important *Fusarium* mycotoxins are trichothecenes, zearalenone (ZEN) and fumonisins (FUM) [40]. ZEN occurs more commonly than its metabolites. FUM group is represented by fumonisin B_1_ (FB_1_), B_2_ (FB_2_), and B_3_ (FB_3_), FB_1_ being the most abundant member [28]. Trichothecenes can be divided into four types (A, B, C, D); the concerns regarding type A and type B trichothecenes are higher due to their higher toxicity and occurrence in crops [41,42]. Known mycotoxins that belong to type A trichothecenes are T-2/HT-2 toxin, diacetoxyscirpenol (DAS) and neosolaniol (NEO). Among the type A trichothecenes, T-2 toxin is the most toxic mycotoxin regardless of the exposed animal species, is soluble in non-polar solvents (e.g., ethyl acetate and diethyl ether) and is rapidly metabolised to HT-2 toxin [43,44,45]. Known mycotoxins that belong to type B trichothecenes are DON, nivalenol (NIV), fusarenon X (FX) and fusaric acid (FA). Among the type B trichothecenes, worldwide [41,46] DON is the most commonly found mycotoxin in cereal grains.

Besides the “traditional” *Fusarium* mycotoxins described above, *Fusarium* species produce other metabolites called “emerging” mycotoxins such as fusaproliferin (FUS), beauvericin (BEA), enniatins (ENNs), and moniliformin (MON) [47]. Furthermore, *Fusarium* mycotoxins can occur as plant-derived derivatives which are often not detectable during routine mycotoxin analyses and, therefore, called “masked” mycotoxins, after having been biologically modified by plant defense mechanisms after crop infection [20,46]. The most commonly-detected masked mycotoxin conjugates are β-linked glucose-conjugates of trichothecenes: DON-3-glucoside (DON3Glc), nivalenol-3-glucoside (NIV3Glc), HT-2 glucoside (HT2Glc), and ZEN-14-glucoside (ZEN14Glc) [48]. Masked mycotoxins are derived from conjugation reactions following a glucosidation reaction, but can also involve glucuronidation or sulfatation (Phase II of plant metabolism), and are usually less harmful than the parent mycotoxins [46,49]. However, masked forms might be “reactivated” during animal digestion by the action of gut microbiota, which may cleave the polar group and consequently liberate the parent toxin [46]. The concept of toxin reactivation has been confirmed for DON3Glc and NIV3Glc in rats [50,51] and for DON3Glc and ZEN14Glc in pigs [52,53]. To avoid confusion [54], one should not only distinguish free mycotoxins from masked mycotoxins, but also from matrix-associated and other modified mycotoxins. To further emphasize the distinction, acetylated derivatives of DON such as 15-acetyl DON (15AcDON) and 3-acetyl DON (3AcDON) are fungal metabolites (free mycotoxins). These toxins are commonly detected along with DON in feedstuffs and animal feeds [16]. In other words, mycotoxins can be present in many forms.

In Europe, AFB_1_ is the only mycotoxin regulated by the Directive 2002/32/EC of the European Parliament and of the Council of 7 May 2002 on undesirable substances; for fish species the maximum allowed concentration in feed materials is 20 µg/kg (ppb), and for complete feed is 10 ppb (Appendix A) [55]. For other mycotoxins, including important *Fusarium* mycotoxins such as DON, ZEN, T-2 and HT-2 toxin, FB_1_ and FB_2_, the EC has established only recommended limits for their presence in feedstuffs and feed (Appendix A) [56,57,58]. Among these recommended limits only those for FB_1_ and FB_2_ refer directly to fish species. In addition, European Commission (EC) regulations/recommendations are based on the occurrence of a single mycotoxin, although feeds are usually contaminated by numerous mycotoxins simultaneously that might, in some instances, result in synergistic effects [59].

The present study aims to extrapolate from a large dataset and thus highlight the potential threat of mycotoxins to European aquaculture by (a) unravelling mycotoxins patterns in both, fish feeds and in the commonly used plant-based feed ingredients: wheat, corn and soybean meal; (b) updating the current state of knowledge on the effects of DON and the risk of DON exposure on important farmed fish species; (c) predicting the effects of DON on fish performance; and (d) providing practical advice for fish farmers and fish feed manufacturers.

## 2. Results

### 2.1. Survey of Feed Ingredients and Aquafeeds

#### 2.1.1. Wheat

##### Wheat as an Ingredient in Aquafeeds

A total of 266.1 Mt of wheat was produced in 2019 in Europe, where wheat is the main cereal crop [60,61]. Wheat productivity, as for other crops, is dependent upon an optimal range of both temperature and precipitation [62]. Based on the predictions of several mathematical models, it is estimated that the production of wheat will fall by 6% for each °C of further temperature rise and thus future wheat productivity could become uncertain [63]. Also, fungal growth is dependent on environmental factors such as temperature, but also pH, water availability, nutrients and light and, therefore, it is rational to assume that climate change will affect wheat production through a direct effect on fungal and mycotoxin presence [64]. The majority of wheat produced is milled and destined for human consumption, while only a portion of the total production and almost all of the milling by-product (wheat bran) is used as an ingredient in feeds for both terrestrial animals and fish [65]. Fungi mainly grow in the outer part of the kernels, and consequently, the relative concentration of mycotoxins is higher in wheat bran [66]. The essential amino pattern in wheat and its by-products is unbalanced, so that these feed ingredients are primarily incorporated in aquafeeds as the main starch source, to function as binders that improve water stability of the pellets [65]. These nutritional characteristics are the main reason that traditional formulation software restricts its inclusions in fish feed formulations, especially for carnivorous species [67]. For instance, the average inclusion of wheat was reported as low as 9.9% for salmon in Norway [8] or 10.6% for trout feeds in France, Greece, Denmark, Norway and UK [61]. Also for marine species farmed in Europe the average inclusion is low at 7.5% while, in contrast, in feeds for herbivorous/omnivorous tilapia, the average wheat inclusion can be as high as 19.9% [61] (Appendix A). Thus, wheat inclusion rate varies within feeds for different fish species.

##### Mycotoxins in Wheat

From our analysis of *n* = 857 wheat samples from European countries, 42 distinct mycotoxins were retrieved, including regulated toxins, mycotoxins with guidance levels, masked as well as emerging mycotoxins (Table 1). Interestingly, 80% of the tested samples were positive for at least one mycotoxin, and in 63% of the analysed samples more than one mycotoxin was found. Average mycotoxin co-occurrence was four, and the maximum number of different toxins present in one sample was 14. Mycotoxin co-occurrence in wheat has been reported in previously published surveys [68,69,70,71] although the figures cannot be directly compared since only a few toxins were analysed, and incidence of co-occurrence is presented only for either animal feed samples [18], or for all matrixes analysed [72]. Finally, data from 8 years of field surveys revealed a co-occurrence of DON and ZEN, and between DON/ZEN and their modified forms in cultivated wheat in the Netherlands [73].

Our analysis showed DON to be the most frequently reported toxin, detected in 41% of the samples (348/857 positive samples), followed by FB_1_ (27%) and FX (23%). Average and maximum values of toxin contamination for the analysed toxins are given in Table 1. Average DON contamination was 470 µg/kg, with 8872 µg/kg being the highest level of DON detected in a sample from Lithuania in 2017. The Lithuanian sample was the only sample that exceeded the critical limit in cereals recommended by EC, currently set at 8000 µg/kg (Appendix A). Highly comparable to our findings, a recent report on the occurrence of DON in wheat samples from Europe [23] mentioned an average contamination level of 418 μg/kg with a maximum of 6219 μg/kg. The most extreme value of DON so far reported for wheat/wheat bran was 49000 μg/kg found in Central Europe, with an average contamination of 848 μg/kg [18]. Furthermore, DON characterized as the most frequent mycotoxin in cultivated wheat in The Netherlands, which occurred on average in 54% of the samples with a mean DON contamination of 228 μg/kg [73]. These data come from 8 years of field surveys and revealed that DON contamination in wheat was mainly affected by year and region. In contrast, agronomic practices (fungicides against *Fusarium* spp, crop rotation, resistant wheat cultivars) did not have an influence on DON contamination in wheat. Most commonly, DON levels in wheat appear to be governed by climatic conditions and below the critical limit.

Other important *Fusarium* toxins, like ZEN and T-2 toxin in wheat, were detected in 5% and 7% of the cases, respectively, with only one sample containing 551 µg/kg T-2 toxin, slightly above the critical limit set by the EC (Appendix A). The emerging mycotoxins BEA and MON were present in only 1% of the analysed samples with a maximum contamination level of 14 and 24 µg/kg, respectively. Although ENNs have been reported as the most frequent toxins in Romanian wheat grains and flour samples [74], in our current study wheat samples were not analysed for ENNs. Also, masked mycotoxin DON3Gluc (13%) was found in 53 wheat samples harvested in Serbia, although at low contamination levels from 17 to 83 µg/kg [70]. We detected DON3Gluc in only 7% of the samples, with a maximum value of 1072 µg/kg, suggesting “traditional” DON being most frequent in wheat.

#### 2.1.2. Corn as an Ingredient in Aquafeeds

The European production of corn reached 132.8 Mt in 2019, corresponding to 11.6% of the total corn produced that year globally [60]. Corn gluten meal (CGM) is a product derived from the wet-milling processing of corn, with an adequate crude protein content of 60% which is highly digestible. Therefore, it is often used as a protein source in fish diets, although due to its deficiency in lysine, diets are usually supplemented with synthetic amino acids or combined with other protein sources to meet the animals’ nutritional requirements [10]. Corn itself can be included in the diets of omnivorous species [65] such as Nile tilapia (*Oreochromis niloticus*) and common carp (*Cyprinus carpio*) at average inclusion levels of 27–30% [61]. For carnivorous species (trout, salmon) and marine species (European sea bass, *Dicentrarchus labrax*, gilthead sea bream, *Sparus aurata*), CGM is often used [10,65]. Its inclusion rate can, therefore, be lower in diets for rainbow trout (*Oncorhynchus mykiss*) at 7.5% and for sea bass and sea bream at 8.8% [61,65]. Corn inclusion rate in aquafeed therefore varies with the exact product (corn versus corn gluten meal) and with the fish species.

##### Mycotoxins in Corn

From our analyses for regulated toxins, mycotoxins with guidance levels, masked and emerging mycotoxins based on *n* = 725 corn samples from Europe we could reveal the presence of 40 different mycotoxins (Table 1). According to the survey results, at least one mycotoxin was found in 95% of all analysed corn samples, whereas in the majority (88%) of samples multiple mycotoxins were detected. The highest number of mycotoxins that were simultaneously present in a single corn sample was 17, while the average number of mycotoxins co-occurring in corn was 6.

A comparison of our mycotoxin co-occurrence dataset with other studies might not be directly informative due to the inconsistency of the available information presented in the literature, as discussed previously for the wheat data. Yet, the presence of numerous mycotoxins is a phenomenon well described in literature [19,20,23,75,76]. *Fusarium* toxins appear to be among the most frequent mycotoxins present in corn (Table 1). Among the *Fusarium* mycotoxins with a guidance level, FB_1_ was found in 70% of the samples, followed by FB_2_ (54%), DON (47%), ZEN (16%) and T-2 toxin (14%). Data analysis also showed a high frequency of *Fusarium* mycotoxins without any regulated or recommended limit; FA (67%), FB_3_ (41%), 15AcDON (20%), 3AcDON (14%) and FX (10%) and of *Fusarium* emerging mycotoxin MON (10%). Besides the *Fusarium* mycotoxins, a *Penicillium*-derived mycotoxin, roquefortine C was detected in 10% of the corn samples. In comparison, a three-year survey of corn samples in Europe [18] for aflatoxins (AFLAs) (31%), ZEN (30%), DON (72%), FUM (60%) and OTA (10%) estimated high frequencies of FUM, DON and ZEN similar to our observations. A recent survey [76] combined the yearly presence of mycotoxins in corn harvested in Serbia between 2012–2015 with meteorological data and thus linked observed differences in mycotoxin patterns to different weather conditions. For instance, the high occurrence of AFLAs in 2012 could be related to the prolonged drought reported that year and the high occurrence of DON and ZEN in 2014 could be linked to extreme precipitation. Regardless of the year and weather conditions, FUM were dominating (76–100%) in the corn samples. Unfortunately, in our study, it was not possible to correlate mycotoxins with meteorological data since our database was generated from samples originating from various locations in Europe.

In our survey of corn samples, AFB_1_ (4%) and OTA (9%) did not often occur, although in five cases AFB_1_ was above the regulated limit of 20 µg/kg, and in one sample OTA exceeded the recommended limit of 250 µg/kg. Of interest, a predictive model on the occurrence of AFB_1_ under a climate scenario of 2 °C increase due to global warming within the next 100 years shows that this toxin will become a serious food and feed safety concern in corn, even in temperate areas like Europe [77]. In our survey of corn samples, the maximum level for DON was 10,020 µg/kg and the maximum level for ZEN was 1282 µg/kg. Others have reported values for DON = 26,121 µg/kg and ZEN = 849 µg/kg [18], or DON = 4000 µg/kg and ZEN = 10,000 µg/kg [20] or DON = 19,180 µg/kg and ZEN = 8888 µg/kg [23]. In all cases, the maximum DON level exceeded the EC guidance level of 8000 µg/kg. In our database, only three samples were detected with a DON level above this limit, whereas for ZEN all samples were below the EC guidance level (<2000 µg/kg). Similarly, levels of the most frequently occurring toxin in our corn samples, FB_1_ were below the EC recommendation (60,000 µg/kg). Occurrence of T-2 and HT-2 toxin were collectively examined with 9% positive samples above the guidance of 500 µg/kg and a maximum level of 3340 µg/kg. Other frequently detected toxins cannot be assessed for risk levels since there is not regulatory or guidance limit by the EC. Maximum contamination levels of FA (4327 µg/kg), FB_3_ (3203 µg/kg), 15AcDON (1667 µg/kg), 3AcDON (406 µg/kg) and FX (604 µg/kg) detected in our samples are difficult to compare because other surveys have not analysed corn for these toxins. Only [20] discussed the presence of the emerging toxin MON in corn samples from Southern Europe but reported generally low concentrations (<100 µg/kg). The same study reported the highest MON values (400 µg/kg) in South Africa, whilst our dataset showed a maximum of 1103 µg/kg with an average MON contamination of 171 µg/kg in European corn samples, which is relatively low.

#### 2.1.3. Soybean Meal

##### Soybean Meal as an Ingredient in Aquafeeds

Soybean meal (SBM) is one of the most commonly used plant-protein ingredients to substitute fishmeal in aquafeeds [78], although its inclusion is restricted due to its low crude protein level (48%), limited methionine content, and the presence of anti-nutritive compounds such as saponins [10]. Average values for SBM inclusion have been estimated at 21.3% for salmon diets [8] and, based on extrapolation [61], estimated at 15.5% in trout diets, 19.2% in sea bass/sea bream diets and 13.5% in carp diets. In trout, SBM appears to increase the permeability of the distal intestinal epithelium and limit the capacity of this region to absorb nutrients [79], whereas inclusion of untreated SBM up to 30–45% resulted in histopathological alterations in the intestine, described as reduced numbers of absorptive vacuoles and numbers of goblet cells [80]. Similarly, in Atlantic salmon, inclusion of 30% SBM caused pathological effects in the distal intestine, described as reduced height of tissue folds and reduced vacuolization [81]. In common carp, dietary inclusion of 20% SBM induced intestinal inflammation which diminished after a few weeks of feeding, implying the ability of carp to adapt to SBM ingestion [82]. In marine sea bass, inclusion of 30% SBM in the diet did not adversely affect growth, gut histology, or blood parameters [83]. Also tilapia can tolerate high inclusion levels of SBM, with average inclusion rates of 30.9% (Appendix A) [61]. Tilapia fingerling growth and health do not seem to be compromised by total replacement of fishmeal by SBM (55% inclusion with supplementation of 0.5% L-lysine) [84]. Overall, the effects of SBM inclusion in diets depend on the fish species.

##### Mycotoxins in Soybean Meal

We analysed 139 SBM samples in total for regulated, emerging and masked mycotoxins, in addition to those with a guidance level. Results showed that 33 individual toxins were detected in SBM (Table 1). At least one mycotoxin was detected in 87% of the analysed SBM samples and in the greater portion (75%) of these positive samples more than one mycotoxin occurred. On average, co-occurrence of mycotoxins was four, with a maximum of 12 different mycotoxins. We report higher values than an earlier study of European SBM samples in 2015 [23], which reported 58% positive samples and 32% co-occurrence in (only) 19 SBM samples. Similar to the high (75%) percentage of co-occurrence we report, a study of soya used for animal feed production in Italy also reported 72% of the samples contained at least two mycotoxins [85]. Co-occurrence of several mycotoxins, therefore, appears common.

Of all mycotoxins in SBM, the ones produced by *Fusarium* fungi were the most common (Table 1), with FA being the most represented toxin in our samples (42%). To our knowledge, this is the first study that analysed and reported FA occurrence in SBM. Following FA (42%), we report common occurrence of FB_1_ (26%), T-2 toxin (23%) and FX (12)%. Besides these mycotoxins produced by *Fusarium* fungi, also *Penicillium*/*Aspergillus*-derived mycotoxin OTA (12%) and *Aspergillus*-derived sterigmatocystin toxin (12%) were found frequently. Relatively low sample numbers (11%) were positive for DON, with a maximum contamination level of 543 µg/kg. An earlier study in 2004 reported similarly low occurrence (9.1%, 110 µg/kg) in Serbia, but was based on only 11 analysed samples. In strong contrast, other surveys reported DON as the most prevalent toxin in SBM with a maximum contamination level of 930 µg/kg [23], maximum contamination levels of 714 and 908 µg/kg in samples from Central and Southern Europe [18], or average DON contamination levels of 2600 µg/kg with a maximum of 6400 µg/kg in Italian soya [85]. Despite the inconsistency in DON contamination levels, possibly related to sampling differences (method, geographic location, climatic conditions), contamination levels were always below the EC recommended limit (<8000 µg/kg). In our database, only one SBM sample originating from Germany in 2017 was contaminated with T-2 and HT-2 toxin levels (560 µg/kg) that exceeded the EC guidance value. Overall, SBM showed relatively low contamination levels compared to contamination levels in wheat and corn. SBM is a co-product of oil extraction from soybeans and exposed to high temperatures during the processing step of toasting and perhaps heat treatment helps eliminate mycotoxins from SBM [86]. More extensive screening may reveal more consistent values for DON contamination of SBM in the future.

#### 2.1.4. Probability of Mycotoxin Co-Occurrence in Feedstuffs: The Case of Corn

Our results demonstrate that corn represents a matrix with the highest risk of mycotoxin contamination, but the precise explanation for this is unclear. The reason that corn serves as such a prime host for fungal growth might be related to host genotype [38]. Whereas corn defense systems can respond to fungal pathogens through the expression of defense-related genes, expression of such genes seems to be low in susceptible corn varieties [87]. In general, corn acts as a host to multiple fungi [88] and thus multiple mycotoxin contaminations may prevail in corn fields.

Indeed, previous research confirmed DON occurrence in corn samples to be correlated with other toxins although specific co-occurrence patterns were only hypothesized but not identified [19]. A recent search in literature [89] suggested that DON + FUM had the highest probability (74.4%) of co-occurrence in European corn samples, but also concluded that further research is needed to identify co-occurrence patterns of multiple mycotoxins based on field investigations. In our samples, DON frequently co-occurred with other *Fusarium* mycotoxins; FA (32%), FB_2_ (26%), 15AcDON (19%), ZEN (14%), 3AcDON (12%), DON3Gluc (7%) and FX (6%). A test for significance (Spearman, *p* < 0.05) confirmed a correlation between DON-positive samples and associated toxins with a concentration above the detection limit. A significant moderate correlation (r > 0.5 and *p* < 0.0001) was revealed for the following mycotoxin combinations: DON + 3AcDON (r = 0.57), DON + 15AcDon (r = 0.62), DON + ZEN (r = 0.64). We also investigated the concept of mycotoxin co-occurrence in feedstuffs as the likelihood of association between DON and other toxins (“Toxin X”) detected in corn, and data were expressed as odds ratio (OR). Results from the OR test showed that exposure to specific toxins is associated with at least two times higher odds of DON occurrence (OR > 2 and *p* < 0.05): DON3Gluc, 15-AcDON, NIV, 3AcDON, ZEN, alternariol, roquefortine C, sterigmatocystin, HT-2 toxin, T-2 toxin. The association of DON with the other 39 toxins detected in corn is displayed in Table 2. The toxins in Table 2 are ordered from the highest to the lowest significant OR, followed by the toxins for which the OR was not found significant. Only for OR < 1, the toxins are ordered from the lowest to the highest value because in these cases, when “Toxin X” (BEA, MON, AFB_2_) is present there are fewer odds for the presence of DON. In other words, when “Toxin X” is absent there is a higher risk for the presence of DON.

In corn, DON is more likely to co-occur with other mycotoxins when it is present in its acetylated (3-AcDON, 15-AcDON), modified forms (DON3Gluc). Also, there are higher odds that DON co-occurs with some *Fusarium* toxins (ZEN, T-2 toxin, HT-2 toxin, NIV), while fewer odds with other *Fusarium* toxins (BEA, MON) and aflatoxin B_2_ (AFB_2_). Available data on the *Fusarium* species and their mycotoxins from maize ear rot in Europe are used to discuss our observations. For the correlation of DON with the chemotypes ZEN, NIV and DON, associated forms might occur because they can all be produced by the strains *F. graminearum* and *F. culmorum* [90]. By contrast, the negative association of DON with the following toxins might be because they are produced by different fungi; BEA (*F. subglutinans* and *F. proliferatum*), MON (*F. avenaceum, F. proliferatum* and *F. subglutinans*), AFB_2_ (*A. flavus*) [90]. T-2 and HT-2 toxin are mainly produced by different strains than DON: *F. sporotrichioides, F. acuminatum* [90], although we hypothesize that the positive correlation between these chemotypes could be explained by a positive interaction between their fungi. In general, information about the interactions between individual fungal strains is not always available, and we cannot always expect that observed correlations are an outcome of a similar relationship between the relevant mycotoxin-producing fungi [91]. For example, the latter study found a significant positive correlation between AFB_1_ and FUM levels, but not between the incidences of *A. flavus* and *F. verticillioides*. Thus, mycotoxin production might be driven more by climatic conditions than by the distribution of their corresponding mycotoxin-producing fungi.

#### 2.1.5. Aquafeeds

##### Mycotoxins in Aquafeeds

All feed samples analysed (*n* = 44) were contaminated with at least one mycotoxin (Table 3). A total of 75% of the samples contained more than one mycotoxin simultaneously, and on average a range of 3 to 9 out of a possible total of 24 mycotoxins was found in aquafeed samples. Likewise, another study of aquafeed samples from Asia (*n* = 31) and Europe (*n* = 10) revealed that in 76% of the samples more than one toxin co-occurred [92]. Our data confirm the general observation that animal feed samples often contain multiple mycotoxins (75–100%), especially when more than one plant feed ingredient is included in the diet formulations [19].

The most representative toxins belong to the *Fusarium* group; FA (55%), DON (48%), FB_1_ (36%), FB_2_ (27%) and the masked mycotoxin DON3Gluc (18%). For instance, an *Aspergillus*-produced mycotoxin, verruculogen, was present in only 9% of the samples, but with an average contamination level of 560 µg/kg and maximum contamination 636 µg/kg. None of the previous aquafeed mycotoxin surveys had analysed and thus reported the presence of verruculogen. Surprisingly, information is also lacking for FA even if it was the most frequent toxin in our samples with a maximum concentration of 265 µg/kg. Similarly, the existence of DON3Gluc, FB_2_ and penicillic acid was not previously reported in published data on aquafeed samples. Overall, it was recommended to analyse aquafeed samples for masked mycotoxins like DON-3-glucoside due to their potential to be metabolized to the parent toxin by commensal lactic acid bacteria in the gastrointestinal tract [93].

Typically, DON has been described as the most common mycotoxin in animal feeds [19] and fish feeds [92]. In our study, the average contamination level of DON was 136 µg/kg and the maximum contamination level of DON was 469 µg/kg. Earlier, DON had been identified in commercial aquafeeds with an average contamination of 166 µg/kg and a maximum of 282 µg/kg in 2014 [92]. A pilot survey that included 11 samples of different commercial carp feeds from Central Europe detected ZEN in all samples (average contamination 67.9 µg/kg, maximum 511 µg/kg) and DON in 80% of the samples (average contamination 289 µg/kg, maximum 825 µg/kg) [21]. By contrast, out of the 44 samples in the present study, only one sample was positive to ZEN with a concentration of 348 µg/kg. We also observed that in DON positive samples, FA was present in 62% of the cases, FB_1_ in 48% and DON3Gluc in 24% of the cases. Our findings address, for the first time, DON contamination in aquafeeds along with other toxins. Previous research studies have not evaluated the toxicological effects of these mycotoxin mixtures on different fish species. Even if detected DON3Gluc concentration was low (average 98 µg/kg, maximum 155 µg/kg) it might potentially increase the total bioavailable DON in the intestinal lumen of the animals. Likewise, high levels of FA were not detected (average 41 µg/kg, maximum 265 µg/kg), although in combination with DON it appeared to induce synergetic effects in pigs [94]. Overall, in our European fish feed samples, DON and other mycotoxins with a regulated/guidance value were compliant with the EC limits. Nevertheless, these limits are not customized to fish and importantly do not consider species sensitivities. In the following sections, fish susceptibility to DON will be evaluated based on in vivo dose-response exposure studies and take into account differences in species sensitivities.

### 2.2. Effects of Deoxynivalenol (DON) on Fish Species

As previously mentioned in Section 2.1, mycotoxins are readily present in plant ingredients: corn > wheat > soybean meal and in aquafeeds. In terms of occurrence and toxicity, DON has been characterized as the most high-risk mycotoxin in aquafeeds. Therefore here, by a systematic review we will summarize DON effects on different fish species. In parallel, data were collected in order to quantify the risk of exposure in fish. Finally, by employing a meta-analytical approach, the extent to which DON affects feed intake and growth performance was evaluated. Details on the studies used for this systematic review and meta-analysis are given in Appendix A, respectively.

#### 2.2.1. Systematic Review

Like all trichothecenes, DON binds to ribosomes inducing a “ribotoxic stress response” that activates mitogen-activated protein kinases (MAPKs). The latter are components of a signaling cascade that regulate cellular processes; proliferation, differentiation, stress response and apoptosis [95,96] and mediate inflammatory responses by altering the binding activities of specific transcription factors that lead to induction of cytokine gene expression [97]. Additionally, DON causes oxidative stress in cells by damaging mitochondria function, either by excessive release of free radicals including reactive oxygen species (ROS) which induce lipid peroxidation or by decreasing the activity of antioxidant enzymes [98]. Oxidative stress via the mitochondrial pathway can also induce apoptosis via MAPKs by the caspase-mediated cellular apoptosis pathway [98,99]. Predominantly, rapidly proliferating cells with a high protein turnover such as immune cells, hepatocytes and epithelial cells of the digestive tract are affected by DON [100,101]. Earlier studies in mammals have demonstrated how the mechanism of action of DON affects gut functions (integrity, absorption, immunity), liver functions and the immune system [101,102,103,104,105]. In contrast, earlier studies in fish mainly focused on indirect impacts of DON on productivity, e.g., feed intake, feed efficiency and growth performance [106,107]. Therefore here, when available, we also review the direct biological effects of DON in different fish species. The majority of the studies we reviewed exposed fish to DON through experimental satiation feeding regimes. We will indicate in our systematic review when fish were exposed to DON through restrictive feeding regimes. Also, we will mention if the studies we reviewed exposed fish to “natural” DON (derived from naturally contaminated feed ingredients and other toxins might be present in the aquafeed) or to “pure” DON (extracted and purified to exclude the presence of other toxins). Finally, we will describe the metabolic fate of DON in fish.

##### Salmon

In total, three in vivo studies have been reported that investigated the effects of DON in salmon, and all employed similar experimental conditions; exposure (8 weeks), age (12 months post-smoltification) and source of the toxin (pure DON) [108,109,110]. Reduced growth performance (feed intake and weight gain) was observed in salmon fed the highest DON-containing diet (6000 µg/kg), but not in the low-DON group (2000 µg/kg) [108]. In a follow-up study by [109], more dietary DON doses were used; 0, 500, 1000, 2000, 4000 and 6000 µg/kg. In this case, negative effects on growth performance appeared already in salmon receiving 4000 µg/kg DON; a significant decrease in feed intake was visible after 4 weeks and a reduced condition factor after 3 weeks of exposure. Salmon treated with the highest DON dose (6000 µg/kg) showed reduced weight gain after 3 weeks, and reduced body length and increased relative liver weight after 6 weeks of exposure. After 8 weeks of DON exposure, triglycerides were reduced at 1000 µg/kg, cholesterol, total proteins and albumin, bile acids, packed cell volume at 2000 µg/kg and alkaline phosphatase at 6000 µg/kg.

The most recent study in salmon [110] tested a DON dose of 5500 µg/kg DON against a control treatment. Their findings confirmed impaired salmon performance (reduced feed intake, weight gain, and feed efficiency), and demonstrated for the first time a potential alteration of intestinal integrity and immunity after DON exposure. Specifically, they noted lower relative expression of proteins regulating paracellular permeability between adjacent intestinal epithelial cells, the tight junction proteins (TJPs). Also, an increased relative gene expression of immune markers (suppressors of cytokine signaling, SOCS); SOCS1 (expressed in pyloric caeca and distal intestine) and SOCS2 (expressed in the distal intestine) suggested altered immune regulation to prohibit intestinal damage and inflammation. In all intestinal segments, increased cell proliferation (base on immunohistochemical staining of PCNA, proliferating cell nuclear antigen) was noted in DON-treated salmon, interpreted as a local response to restore intestinal integrity. The total number of goblet cells was unaffected by DON exposure.

##### Rainbow Trout

The first scientific information about the effects of DON on rainbow trout was published in the 1980s [111]. A dose-response exposure study (1000 to 13,000 µg/kg) on juvenile trout for 4 weeks showed that increasing levels of DON resulted in reduced feed intake, weight gain and feed efficiency. Regression analysis suggested that for doses >5000 µg/kg each additional 1000 µg/kg of DON would suppress feed intake by 9% and weight gain by 11%, and for doses >7500 µg/kg each additional 1000 µg/kg of DON would suppress feed efficiency by 6%. In a preliminary experiment as part of the same study, after exposing trout to extremely high DON doses (>20,000 µg/kg) for 4 weeks the authors reported a dramatic drop in feed intake within 5 days and a refusion of pellet ingestion. Of interest, after switching back to feeding non-contaminated diets for four more weeks, feed intake and growth recovered, implying the ability of rainbow trout to adapt to DON, at least after a short-term (4 weeks) exposure.

Surprisingly, no follow-up research was published for 28 years, until a comprehensive article [112] defined rainbow trout as a fish species highly sensitive to DON. The authors showed that increasing levels of natural DON (300, 800, 1400, 2000, 2600 µg/kg) in diets of juvenile rainbow trout for 8 weeks, had a detrimental effect on growth performance, mirroring the effects described earlier [111] even at considerably lower DON doses. At the top of growth performance, exposure to 1400 µg/kg DON significant reduced nitrogen (g/fish) and energy (kJ/fish) retention and their retention efficiencies (%). In addition, body composition analysis of trout fed a contaminated diet with 2600 µg/kg DON showed reduced crude protein content, although no change was observed in the apparent digestibility of crude protein and gross energy. Histological examination of the liver revealed congestion and subcapsular edema with a fibrinous network in rainbow trout exposed to ≥1400 µg/kg DON and multifocal areas fatty infiltration and phenotypically altered hepatocytes (pyknotic and karyolytic) in trout exposed to 2600 µg/kg DON. Moreover, to explore DON effects not related to differences in feed intake, authors employed an additional treatment; fish pair-fed the control diet the same amount of feed consumed by fish fed the highest DON dose (2600 µg/kg). Fish fed the DON diet showed significantly reduced growth rate (thermal growth coefficient; TGC), feed efficiency, protein and energy utilization efficiencies and whole body crude protein compared to the fish pair-fed the control diet. This observation suggests that reduced growth performance is not fully attributed to a reduced feed intake, but also metabolic disturbances related to the direct effects of DON on the cellular level. In contrast to [112], in other experiments pair-feeding showed that suppressed weight gain in fish fed DON-contaminated diets might arise from depressed feed intake [113,114]. However, the studies differed in trout size (~24 g [112] and ~103 g [114]). Apart from the indirect effects on feed intake, DON toxicity may be age-dependent, with young trout being more vulnerable to metabolic effects of DON. Following the study in 2011 [112], later studies confirmed a significant reduction in feed intake (≥4100 µg/kg) upon offering diets with increasing levels of natural DON (500, 4100, 5900 µg/kg) [114] and (≥3100 µg/kg) by testing diets with 100, 3100, and 6400 µg/kg natural DON [113]. Moreover, the latter study in a sub-experiment measured reduced feed intake at the two tested DON doses (3300 µg/kg natural DON and 3800 µg/kg pure DON).

Subsequently, follow-up experiments on rainbow trout followed that investigated, next to the effects of DON on performance, nitrogen and energy balances and carcass composition, effects of a commercial anti-mycotoxin additive [115], potential synergy among *Fusarium* toxins present in naturally contaminated trout feeds [116], the impact of diet composition on detoxification capacity, and species sensitivity in a comparison with tilapia [117]. Trout fingerings (initial weight; 1.8 g) exposed to natural DON for 12 weeks showed reduced feed intake, weight gain, TGC, reduced nitrogen retention efficiency (≥1000 µg/kg), and reduced retained nitrogen (≥1500 µg/kg) [115]. None of these effects could be reversed by the inclusion of a commercial feed additive, suggesting that anti-mycotoxin products developed for homeothermic species might not be as effective in cold-blooded species, such as trout. In another study [116], diets with graded levels of pure DON (0, 700, 1400 and 2100 µg/kg) or natural DON (0, 2100, 4100 and 5900 µg/kg) were offered to rainbow trout (initial weight; 50.3 g) for a period of 8 weeks. Regardless of the DON source (pure/natural), deleterious effects were present, and similar trends of reduced retained nitrogen, recovered energy, nitrogen retention efficiency (≥2100 µg/kg pure/natural DON), and energy retention efficiency (>2100 µg/kg natural DON) were found. The same study [116] was the first to use histological examination to show harmful effects of DON on the gastrointestinal tract after feeding 2100 µg/kg pure or 5900 µg/kg natural DON. Last but not least, the most recent work of these authors [117] investigated if increased levels of digestible starch (12% vs. 24%) in rainbow trout diets contaminated with 100, 700 and 1300 µg/kg natural DON could help enhance DON detoxification to deoxynivalenol-glucuronide (DON-GlcA) via increased glucuronidation capacity. This did not seem to be the case because, regardless of the starch level, rainbow trout exhibited impaired growth performance, disturbances in nitrogen and energy balances and carcass composition, suggesting that the higher supply of carbohydrates from starch, which presumably increases the hepatic glycogen content, did not directly lead to DON detoxification.

Further studies had also confirmed the impact of DON on rainbow trout productivity; either by using low DON doses (1100 and 2700 µg/kg) [118] or high (4700 and 11,400 µg/kg) [119]. Notably, the latter study provided new insights into the direct effects of DON by measuring proteolytic enzyme activity and relevant gene expression in the head kidney, liver, brain and gastrointestinal tract. Experimental DON doses of 4700 and 11,400 µg/kg indeed affected the activities of proteolytic enzymes (pepsin, trypsin and chymotrypsin), although it remained unclear if the observed changes in enzyme activity were directly related to the toxin itself or a result of reduced feed intake. Surprisingly, gene expression of the neuropeptide Y precursor (*npy*) in the brain was up-regulated for doses ≥4700 µg/kg DON, whereas the opposite would have been expected for this appetite-stimulating precursor. Less surprising maybe, another gene in the brain of which the expression is also related to feed intake and growth control (growth hormone-releasing hormone/pituitary adenylate cyclase-activating polypeptide PACAP; *adcyap1a*) was down-regulated. Also in the liver, expression of genes related to growth control (insulin-like growth factors; *igf1, igf2*) were down-regulated. Finally, some other studies addressed the effects of DON on health, immune function and oxidative stress [120,121,122]. When 1-year-old trout were exposed for 23 days to ~2000 µg/kg DON, plasma biochemical parameters; glucose, cholesterol and ammonia were decreased [120], pro-inflammatory cytokine TNF-α in the head kidney was up-regulated [121] and altered activities of antioxidant enzymes were observed [122]. Overall, the sensitivity of rainbow trout productivity to DON is well defined, although further research is needed to explore the direct mechanism of action of the toxin in this species.

##### Carp

Globally, carp is the most important fish species in terms of total mass production, with grass carp (*Ctenopharyngodon idellus*), silver carp (*Hypophthalmichthys molitrix*) and common carp (*Cyprinus carpio*) listed as first, second and fourth in the list of most intensively farmed fish species in 2018 [123]. Contrary to other species, DON research in carp did not focus mainly on performance but rather targeted its mechanisms of action at the cellular level, and DON effects on health.

A series of studies mostly performed by Pietsch and colleagues in common carp [124,125,126,127] investigated the effects of pure DON on immunity, oxidative stress and liver health. Feeding low doses of DON (352, 619 or 953 µg/kg) for 6 weeks [124], led to increased oxidative stress in several tissues (953 µg/kg dose). As also described for trout [112], fat aggregation in hepatocytes was observed at DON levels ≥ 619 µg/kg, assumed to be a result of the ribotoxic effect of DON on the synthesis of protein-lipid transporters (lipoproteins) [128]. Concentrations of serum protein (albumin) in carp were reduced at DON levels of 619 and 953 µg/kg [124]. Taken together, this implies a negative role of DON on nutrient metabolism. Potentially, DON affects also anaerobic metabolism since the activity of lactate dehydrogenase (LDH) varied in different tissues of DON-exposed carp. For instance, LDH activity increased in head and trunk kidney (≥352 µg/kg), decreased in muscle (953 µg/kg), but LDH activity and consequently lactate concentration increased in serum (953 µg/kg), indicating activation of gluconeogenesis to maintain glucose levels. An additional study measured reduced cell viability and immune function of unstimulated or bacterial lipopolysaccharide (LPS)-stimulated leucocytes derived from the head kidney [125], indicative of cytotoxic effects of DON on immune cells.

DON might have immunostimulatory or immunosuppressive properties, depending on dose, frequency and duration of the exposure, as shown in mammals [129]. Thus, DON studies in carp [126,127] also evaluated duration of exposure to DON after acute (7, 14 days) and sub-chronic (26, 54 days) exposure. Short-term (acute) exposure to 953 µg/kg DON resulted in activation of pro-inflammatory cytokines and anti-inflammatory cytokines. Reduced ROS production, and increased nitric oxide (NO) production in trunk kidney leucocytes after LPS stimulation confirmed a potential immunostimulatory capacity of DON. Longer-term (sub-chronic) exposure resulted in increased mRNA expression of immune-relevant genes in the trunk kidney, while in other organs mRNA expression levels of the same genes returned to the basal levels. Thus, sub-chronic (26 days) exposure to DON appeared to lead to pro-inflammatory responses and to anti-inflammatory responses, to prevent damage from permanent inflammation. Using the same experimental set-up (control vs. 953 µg/kg DON) [127], measuring liver enzyme activities and histological changes indicated a suppression with time of the biotransformation and antioxidative capacity influenced by exposure to DON.

Two more studies investigated the effect of pure DON on oxidative stress [130,131] in common carp. Dietary application of 5960 µg DON per kg feed for 4 weeks did not impair lipid peroxidation in the hepatopancreas [130]. A single, high (1750 µg DON /kg body weight) oral dose given by gavage [131] equivalent to 200,000 µg DON/kg of feed aimed to evaluate short-term (1-day experiment; sampling at 8, 16 and 24 h) responses that could reveal potential DON effects on lipid peroxidation and parameters of the glutathione redox system in the liver. As mentioned above, DON research in carp often focused on mechanisms of action and effects on growth performance were not studied [130,131], or showed no significant effect of DON [124,125,126,127]. Because these studies applied restricted feeding protocols rather than satiation feeding, DON effects on the growth performance of common carp may not be fully conclusive. Notably, juvenile grass carp fed with a DON level of ≥636 μg/kg [132,133,134] showed poor growth performance and body malformation. Finally, there is one study that referred to increased mortality (16.7%, twice higher than the control) associated with exposure to DON (5960 µg/kg) of common carp [130].

DON research on grass carp also focused on unravelling the mechanism of action of the toxin, by addressing effects on oxidative stress and cell apoptosis, and new information was generated on the effects on gut and gill integrity. Investigations on juvenile grass carp [132,133,134] fed until satiation on diets with graded levels of pure DON (27, 318, 636, 922, 1243 and 1515 µg/kg) for 60 days, reported oxidative damage in the intestine after feeding ≥ 318 µg/kg and reported down-regulation of mRNA levels coding for antioxidant enzymes. In addition, for DON doses ≥636 µg/kg, increased lipid and protein peroxidation in grass carp intestine were noted. Intestinal tissue damage was also confirmed at the molecular level by detecting decreased relative mRNA expression of barrier-forming TJPs, indicating impaired gut integrity already at relatively low doses of 318 µg/kg DON (see Appendix A). Following a 60-day growth experiment, grass carp were challenged with *Aeromonas hydrophila* to investigate the effects of DON on intestinal immune function [133]. At doses ≥636 µg/kg, DON exposure impaired innate and adaptive immune responses in the intestine.

##### Zebrafish

Zebrafish (*Danio rerio*) is a well-recognized animal model species for human research and now more frequently is also highlighted as an animal model for other fish species, for example to investigate host–microbe immune interactions and fish health [135] and investigate toxicological effects of mycotoxins in vitro [136]. Indeed, zebrafish could represent an ideal animal model to study biological effects of DON on fish. Surprisingly, we could find only one in vivo study on the toxicity of DON in zebrafish [137]. In this study, although the application of increasing concentrations of 0, 100, 500, 1500, 2000 and 3000 µg/kg pure DON for 45 days to zebrafish (30 days post-hatch) using a restrictive feeding regime showed no effects on growth performance, other effects on sensitive endpoints in biotransformation, oxidative stress, behaviour and reproduction were described. Fecundity, measured as the mean number of eggs produced by individual females, was increased in zebrafish fed with DON 1500 µg/kg, but decreased in zebrafish fed the highest DON dose (3000 µg/kg). To the best of our knowledge, the effects of DON on the fecundity of fish had not been reported before. Effects of DON on behaviour were also examined. A trend for higher swimming activity was found in offspring of zebrafish parents that had been fed the highest DON dose. Nonetheless, freshly fertilized embryos (96, 100 and 120 h) treated with DON (0.01–100 mM) showed no behavioural alterations related to locomotion [138]. No matter what, the first results that come from this single study are sufficiently interesting to warrant further examination of the effects of DON on fish biology using the zebrafish as animal model.

##### Tilapia

Although Nile tilapia (*Oreochromis niloticus*) is the third most important fish species in term of aquaculture, with an annual production of 4.5 million Mt in 2018 [123], research efforts into the effects of DON have not been proportional. The relative lack of effort could be related to Nile tilapia primarily being cultivated in tropical and subtropical areas [139] while DON is the main contaminant in crops present in temperate regions. Indeed, there have been more research efforts on the threats posed by AFB_1_, which is one of the most prevalent mycotoxins in tropical latitudes. To date, only two studies published the effects of natural DON on tilapia [117,140]. In the first study, red tilapia (*Oreochromis niloticus* × *O. mossambicus*) fingerlings were exposed for eight weeks to graded low doses of DON (70, 310, 500, 920 and 1150 µg/kg) along with exposure to ZEN (10, 90, 210, 370 and 980 µg/kg) [140]. Because in this study ZEN levels were relatively high, interpretation of the results is more difficult due to confounding effects of the combined exposure. Consumption of increasing doses of both, DON and ZEN led to a significant linear decrease in growth performance measured as feed intake, weight gain, feed efficiency and thermal daily growth coefficients. Furthermore, the ingestion of highly contaminated diets was linked with either linear or quadratic increase in the percentage of mortalities; an endpoint that had not been reported earlier in studies on DON in fish. Despite the increase in mortality, although lesions were observed in some mycotoxin-treated fish, no significant histopathological alteration in the liver was found and no effects were noted in hematological and biochemical parameters in the blood. In the second study, tilapia were exposed to graded levels of corn naturally contaminated with DON [117]. Exposure of Nile tilapia fingerlings to either a low-starch (12%) or high-starch (24%) diet containing graded levels of natural DON (100, 700 and 1300 µg/kg) and fed until satiation for 10 weeks did not lead to any significant changes in growth performance. Overall, studies on the effects of DON in tilapia have been few and inconclusive.

##### Catfish

The effects of DON on channel catfish (*Ictalurus punctatus*) have been investigated by only a single study in fingerlings using the following doses 0, 3300, 5500, 7700 and 8800 µg/kg [141]. After feeding high doses of DON for 7 weeks, these catfish did not experience negative effects on growth performance such as either impaired weight gain or reduced feed efficiency. In fact, and surprisingly, feed conversion in catfish fed with a high dose of DON (8800 µg/kg) was more efficient than in catfish fed with a low dose of DON (3300 µg/kg). Even more surprising, DON seemed to have a protective role against bacterial infection with *Edwardsiella ictaluri* because catfish fed with high doses of DON (>5500 µg/kg) showed reduced mortality after challenge. Of interest, in an early study of digesta of nine different freshwater fish species (sampled in their natural habitat) on the presence of microbes having the ability to transform trichothecenes to less toxic forms [142], there was one catfish species (*Ameiurus nebulosus*) that stood out from the rest for having a microbial community (culture C133) able to completely transform DON to its less toxic metabolite de-epoxy-deoxynivalenol (DOM-1) after incubation for 96 h at 15 °C. Catfish are omnivorous fish species that naturally feed on plant sources, which could imply they strategically developed through an evolutionary process mycotoxin-transforming microorganisms to help detoxify plant toxins. Undoubtedly, catfish species appear highly tolerant to DON.

##### Metabolic Fate of DON

Toxicity of DON can be reduced via biotransformation of DON to DOM-1 by anaerobic bacteria in the rumen or intestine, or via oxidation to 3-keto DON along with isomerization to 3-epi DON (3-β-hydroxy) by aerobic bacteria [143]. After intestinal absorption, DON can be metabolised in the liver by conjugation mainly to glucuronic acid, sulfate or sulfonate resulting in more hydrophilic and less toxic forms that can be excreted by the animal’s body [144]. Also in fish, glucuronidation can transform DON to DON-3-glucuronide (DON-3-GlcA), at least in in vitro studies of liver microsomes of carp and trout [145]. An in vivo experiment with rainbow trout confirmed metabolisation of DON to the less toxic DON-3-sulfate, possibly explaining the absence of clinical signs with high doses of DON [119]. In general, also in fish, DON and its metabolites are readily excreted via the bile and thereby finally via the faeces [146]. Studies have shown almost negligible accumulation of DON in the muscle of salmon [108,147], carp [125] and gilthead sea bream [147], indicating little risk to humans after consumption of farmed fish fillet. Yet, although crucial for more detailed understandings of the effects of DON on different fish species and potential detoxification strategies, research on the toxicokinetics of DON and its metabolic fate in fish remain scarce.

#### 2.2.2. Quantifying the Risk of DON Exposure in Fish

CC5 values are critical concentrations that affect 5% of a (fish) population. Probabilities and distributions of the estimated CC5 values are displayed in Figure 1 as log 10[concentration of the toxin] with kernel density and box plots. Our risk assessment of DON in fish feed, performed on a large number of fish species (*n* = 146), indicated CC5 values of 43–79.4 µg/kg (mean 59 µg/kg). A previous risk assessment of DON in fish feed based on 39 data points [148], predicted more variable and higher CC5 values of 23.8–272.3 µg/kg (mean 114.8 µg/kg). In an attempt to gain more detailed information on species-specific sensitivity, we estimated CC5 values for three subgroups; rainbow trout (*n* = 56), salmonids (*n* = 67) and all fish species excluding rainbow trout (*n* = 90). This approach led to a threshold for DON in fish feed for only rainbow trout of 43.7 µg/kg (24–75.2 µg/kg), lower than the 74.1 µg/kg mean value for all salmonids (45.7–116.3 µg/kg). The exclusion of rainbow trout from the complete dataset led to intermediate CC5 values of 53.9 µg/kg (36.1–79.3 µg/kg). More studies would be needed to generate more data points and extrapolate robust predictions for individual fish species.

#### 2.2.3. A Meta-Analytical Approach

Our systematic review showed that DON can impair feed intake and growth performance in fish, and our risk assessment revealed critical DON thresholds that might threaten 5% of a fish population. In the next section, we describe the results of a meta-analysis aimed to estimate to what extend DON affects feed intake and growth in rainbow trout, and farmed fish in general, using quantitative data from in vivo studies. For most of these studies, it remained unclear whether instances of impaired growth were an outcome of the observed reduced feed intake or related to increased maintenance requirements due to DON effects at cellular level. Thus, correlation between feed intake and growth data was also studied.

##### Effects of Dietary DON on Feed Intake and Growth

The number of in vivo studies found eligible for our meta-analysis (requirement details in Section 4.4) in all fish species was 11 studies, with a total of 63 data points. Data points were coded as control (*n* = 18) or challenged (*n* = 45). Doses in DON-challenged fish ranged from 310 µg/kg to 11412 µg/kg, with a mean of 2575 µg/kg. Control treatments were not always free of DON and ranged from 0 to 300 µg/kg, with a mean of around 71 µg/kg. Duration of DON challenge was on average 56 days (8 weeks). Out of the 11 studies, seven studies (35 data points) referred to rainbow trout, allowing for a separate meta-analysis. In the data subset addressing only rainbow trout, doses in DON-challenged fish (*n* = 25) ranged from 700 to 11,412 µg/kg, with a mean of approximately 3000 µg/kg, and duration of DON challenge was on average 8 weeks. Table 4 summarizes the characteristics of the two datasets used in our meta-analysis. Detailed characteristics of each study (number of experimental animals per treatment etc.) and data on exposure effects on feed intake and growth can be found in Appendix A.

Feed intake and growth data collected in both meta-analyses were converted to relative values compared to their control and expressed as feed intake (% control) and growth (% control). The effect of dietary DON challenge on relative feed intake and growth was assessed by regression analysis. Exponential curves had the most logical fit and explained the greatest degree of variation in effects on feed intake and growth caused by dietary DON intake. Graphs and estimated equations derived from the exponential model are given in Figure 2.

Our results indicate that each additional mg/kg of DON in the aquafeeds leads to an exponential decrease in feed intake (% control) and growth (% control) independent of fish species, and also for trout specifically. The curves in Figure 2 show a rapid and exponential decline in relative feed intake and growth already for low doses of DON, followed by a slower decline at higher doses. These results indicate a more severe impact on feed intake and growth at low DON doses, while at higher doses the impact will level off. The most striking result from our analysis is that already at doses below the EC recommendation limit (5000 µg/kg) there are adverse effects on feed intake and growth. Fish exposed to a diet with 5000 µg/kg DON show reduced feed intake of only 52% of that of the control group and reduced growth of 39% of the control fish (Figure 2). Even stronger for rainbow trout, fish exposed to a diet with 5000 µg/kg DON showed a predicted feed intake of only 43% of control values and predicted growth of only 36% of the control. It is relevant that the values for feed intake and growth are more acute predictions for trout, as can be observed by comparing the exponents in the equations (Figure 2). In general, with each additional mg/kg of DON, fish show a decline in feed intake of 13.2%, whereas rainbow trout show a stronger decline in feed intake of 18.8%. Similarly, with each additional mg/kg of DON, fish show a decline of 16.5% in growth, whereas rainbow trout show a stronger decline of growth of 20%. Taken together, our results suggest that the current EC recommendation limit might not be sufficiently low to guarantee optimal feed intake and growth in farmed fish species exposed to DON. Our results also suggest that rainbow trout is relatively sensitive to DON.

##### Different Types of DON in Rainbow Trout: Natural vs. Pure

In the data subset addressing experimental studies in rainbow trout (*n* = 35), DON challenge by experimental diet could be the result of two different contamination routes. In most of the cases (*n* = 25), DON was added to the diets in natural form by including naturally contaminated plant-based ingredients. Fewer studies (*n* = 10) investigated the impact of DON by testing pure DON purchased as a commercially available powder. Although in theory this could affect outcomes, a recent comparison of natural and pure DON (2100 µg/kg) exposure of rainbow trout found no difference between these two contamination routes [116]. Further research into this comparison is complicated by the challenge to formulate comparable diets containing identical levels of natural and pure DON.

Regression analysis showed that regardless of the dietary source of DON (natural/pure), feed intake and growth (% control) of trout decreased exponentially with each mg/kg of DON added to the feed (Figure 3). The regression coefficients for natural and pure DON, however, were highly significantly different (*p* < 0.0001). The decline in feed intake for natural DON (22.1%) was much steeper than for pure DON (12.9%). Likewise, the decline in growth was much steeper for natural DON (26%) than for pure DON (15.7%). These findings strongly suggest that feed naturally contaminated with DON has a more severe impact on feed intake and growth of rainbow trout than feeding with contaminations of pure DON. As discussed above, natural DON is derived from naturally contaminated plant ingredients and usually co-exists with other toxins. In growing pigs, a meta-analysis of effects of individual mycotoxins showed a reduction of feed intake (14%) and growth (17%), but much larger reductions after exposure to multiple mycotoxins of 42% for feed intake and 45% for growth [149]. We support the hypothesis that also in fish the occurrence of multiple mycotoxins might lead to synergisms that could explain more severe effects of aquafeed contaminated with natural DON, in comparison to effects in studies using aquafeeds with pure DON.

##### Relationship between Feed Intake and Growth

To date, little attention has been paid to the potential interference between reduced feed intake caused by DON and observed reductions in growth in fish. Only a few studies in rainbow trout added to their experimental design an additional control treatment, in which fish received the same amount of feed consumed by the group challenged with the highest dose of DON (pair-fed). One study showed significantly impaired growth in the DON treated group against the control [112], but other studies [113,114] did not find significant effects. The outcomes from these pair-fed investigations, therefore, are not fully conclusive either. For that reason, we used the data collected in our meta-analysis to further explore the correlation between feed intake and growth response. Regression analysis showed a linear relationship between relative feed intake (% control) and growth (% control) for all fish species, including trout only (Figure 4). According to our model, 94% of the variation in fish growth (98% for trout only) can be explained by the feed intake response. Our data strongly indicate that the impact of DON on fish growth is mostly driven by feed intake. This conclusion, however, is affected by the experimental design of the studies included in our dataset employing satiation feeding strategies, resulting in differences in feed intake. Only if experimental groups are exposed to equal amounts of feed can future experiments aim to unravel the direct effects of DON on fish growth.

## 3. Discussion

We aimed to unravel the profile of mycotoxins present in feed ingredients and thus in fish feeds in Europe, despite the lack of consistent, randomly collected field data. Our study included data from samples submitted by industry, and thus we make the assumption that we cannot fully exclude bias associated with suspicious materials also submitted for analysis. Nonetheless, the current study generated a large set of data and showed patterns related to mycotoxin contamination that are highly relevant to the animal and fish feed industry. We found DON occurrence in 44 European fish feed samples with an average contamination of 136 µg/kg and maximum contamination of 469 µg/kg. So far, comparable data on DON contamination have been derived from much smaller data sets analyzing 11 samples of commercial carp feed (average contamination 289 µg/kg, maximum 825 µg/kg) [21], or 10 samples of commercial fish and shrimp feeds (166 µg/kg and 282 µg/kg) [92]. The much larger number of samples in our data set logically produced more reliable outcomes. Is DON occurrence in feed always detrimental for the fish? Not necessarily. Potentially, high temperatures (>150 °C) during the extrusion process might significantly reduce FUM and ZEA and moderately reduce AFLAs, but extrusion may only slightly reduce contamination with DON in finished feeds [86,150]. For instance, extrusion temperatures above 150 °C only led to a slight reduction of ~20% in DON levels in wheat grits [151]. Overall, complete elimination of mycotoxin is not feasible during feed extrusion and, therefore, prevention of mycotoxin contaminated feeds is of utmost importance for feed manufacturers.

The current survey revealed a risk of association of DON with other *Fusarium* toxins, including emerging and masked mycotoxins. An earlier study based on literature data [59] reported common combinations of different mycotoxins in European cereal samples and addressed their combined risks on different animals, but not fish. Only one study investigated the combined effects of DON with AFB_1_ on the fish cell line BF-2, and combined effects of DON with ZEN on zebrafish larvae [152]. The results implied the existence of effects synergetic between DON + AFB_1_ but antagonistic between DON + ZEN. Future research is needed to investigate similar effects and more diverse combinations of mycotoxins in in vivo feeding experiments. Furthermore, emerging and masked mycotoxins generally are not detectable in routine controls in feed mills, and no regulatory/recommendation limits exist [46]. Thus, feed producers might consider subjecting their raw materials to periodical state-of-the-art mycotoxin analyses performed by external, certified labs to screen the full spectrum of mycotoxins present. Even then, commercial fish feeds when stored under warm (25 °C) and humid conditions (>60% relative humidity) for a month, can release OTA [37]. Thus, to prevent fungal growth and potential mycotoxin contamination after feed production, aquafeed producers and fish farmers have to ensure proper storage conditions.

To the best of our knowledge, this is the first comprehensive study that has attempted to summarize the effects of DON in different fish species using a systematic review approach. Based on our review, we see no evidence for bioaccumulation of DON in fish tissues [108,125,147] and see no reason to raise concerns with respect to consumer health. However, consumption of DON-contaminated feeds by fish, even at levels below the EC recommendation limit (5000 µg/kg), can result in adverse although non-lethal effects on fish such as impaired feed intake, growth performance, immunity, detoxification capacity, and tissue damage and oxidative stress. By collecting all reported adverse effects of DON, our review extended a previous risk assessment [148] and allowed for a new and updated estimation of critical DON levels for rainbow trout, defined as at risk of affecting 5% of a fish population (CC5). This renewed information could have a direct and practical implication for aquafeed producers when designing their mycotoxin management plans.

Undoubtedly the number of studies investigating single effects of DON on farmed fish species has been increasing, but the data have not been collectively used to assess feed intake and growth performance responses. Our meta-analysis provided new insights into aquaculture nutrition that suggest an exponential relation exists between decreases in feed intake and growth response, and increasing levels of DON (mg/kg) in aquafeeds. These adverse effects of DON appear more severe when natural DON is used for feed formulation instead of pure forms of this toxin, as in experimental studies. Other meta-analyses for pigs and poultry similarly showed negative effects on feed intake and growth performance of mycotoxins, including DON [153,154,155]. In summary, our study predicts that the current average contamination of 136 µg DON per kg fish feed leads to 3.5% reduction in feed intake and 3.7% reduction in growth of trout. In a worst-case scenario (maximum DON contamination level of 469 µg/kg), we predict an even greater reduction of 9.9% in growth of trout. *Fusarium* fungal growth, DON contamination and risks of reduced feed intake and growth cannot always be predicted, or ignored. To prevent loss of production therefore, particularly when using diets with high inclusion of plant ingredients for more sensitive species such as rainbow trout, feed manufacturers may consider adding anti-mycotoxin products to aquafeeds and altogether eliminate the risk of mycotoxin exposure.

Another important outcome of our meta-analysis is the attribution of reduced growth performance of DON-challenged fish to reduced feed intake. Feed refusal is a common symptom in animals that have consumed DON and might simply be a response to poor organoleptic characteristics of the contaminated feeds [156] or be considered a natural defence mechanism to minimize risks associated with exposure to the toxin. The mechanism through which DON reduces feed intake may be associated with a direct action on the brain or may be indirect through the secretion of gut hormones [157]. The latter phenomenon remains unexplored in fish, however. In the future, direct effects of DON on fish growth should be studied without confounding effects caused by reduced feed intake. Indeed, to better investigate direct effects of DON on fish growth, future experimental designs need to overcome differences in feed intake between experimental groups by pairwise and equal feeding.

Taken together, mycotoxin contamination is an emerging concern for European aquaculture and requires a multidisciplinary approach. Diverse expertise is needed and, therefore, collaboration and communication of stakeholders from the whole value chain and scientific support from fields such as fish nutrition, toxicology, health and welfare, microbiology, feed processing and technology and plant sciences are crucial. Our findings suggest a strong impact of dietary DON on feed intake and fish growth, and regulatory authorities should reconsider their current DON recommendation limit to ensure economic profitability and protect fish welfare.

## 4. Materials and Methods

### 4.1. Survey

Field surveys regarding mycotoxin occurrence on plant-based ingredients and feeds for aquaculture are not readily available, or at least not at the same extent as those for terrestrial animal feeds and food. Consequently, researchers have no other option than to use inconsistent data from published literature or extrapolate information based on assumptions. This bottleneck has been acknowledged and discussed in a recent publication [148] and aimed to assess the risk for mycotoxin contamination in fish feeds in Europe. To overcome this data gap, we report mycotoxins occurrence data obtained from the database of the Alltech 37+ mycotoxin laboratory (ISO/IEC 17025:2005 accredited), Dunboyne, Ireland. Specifically, the database includes all submitted wheat (*n* = 857), corn (*n* = 725), soybean meal (*n* = 139) and aquafeed (*n* = 44) samples from European countries between 2012–2019. All samples were analysed by the liquid chromatography-tandem mass chromatography (LC-MS/MS) analytical method for detection and quantification of 43 mycotoxins. Mycotoxins’ occurrence was defined considering positive samples, thus samples above the limits of quantification (LOQs) for each mycotoxin. Average and maximum concentration for each toxin were calculated for the positive tested samples. The results are given separately for each matrix. Limits of detection (LODs) and LOQs are available for each toxin and given in Appendix A.

On top of that, the probability of the mycotoxin contamination patterns (DON association with other toxins) is elucidated by applying logistic regression analysis [158]. Results are expressed as odds ratio (OR) with the 95% confidence interval. Statistical Analysis Software (SAS) version 9.4 was used. In our case, the OR represents the odds that DON will occur given a particular presence of “Toxin X”, compared to the odds of DON occurring in the absence of Toxin X. The OR determines if the presence of “Toxin X” is a risk factor for the presence of DON, and expresses the magnitude of the risk (OR = 1: exposure does not affect odds of outcome, OR > 1: exposure associated with higher odds of outcome; OR < 1: exposure associated with lower odds of outcome).

Comparisons of our results with other surveys are briefly discussed since it has been considered that different methods of analysis and different detection limits can generate variability and thus incomparable data. Also, each step of the testing process (sampling, sample preparation and analytical steps) is associated with errors [159]. In the current survey, mycotoxin analyses were performed consistently in terms of the analytical method, although not randomly. Samples had been submitted for analyses by stakeholders associated with the feed industry from European countries and, therefore, generated data might be associated with a degree of bias due to the suspicion of toxin contamination by these stakeholders. In general, taking into account the facts above, mycotoxin results should always be reported with an estimate of the uncertainty.

### 4.2. Systematic Review

A systematic review was conducted by compiling data from scientific articles included research in vivo experiments which evaluated the single effects of DON on fish species. In total, 112 articles were retrieved through database searching on Google Scholar, ScienceDirect, Scopus, Scielo, and PubMed using the keywords: deoxynivalenol (DON/vomitoxin) and fish. Initially, we screened the titles and afterwards the abstracts and by using the PRISMA diagram introduced by [160]. Thus, we selected 28 relevant articles referring to six different species; rainbow trout (13 studies) salmon (3) carp (9) tilapia (2) catfish (1) and zebrafish (1), which were assessed as full-texts and included in the qualitative synthesis summarising the effects of DON per species. The list with the studies used is available in Appendix A.

### 4.3. Risk Assessment

The same articles retrieved from the systematic review were also screened for “no observed adverse effect level” (NOAEL)—the highest dose tested against the control—and the “lowest observed adverse effect level” (LOAEL), the lowest dose tested with a statistically significant effect. We excluded a carp study [161] from the analysis since the level of DON used (200,000 μg/kg) was extremely high and no more realistic LOAEL levels had been included. To assess the risk of DON exposure in 5% of a fish population, we calculated the critical concentration 5% (CC5) based on the mycotoxin levels in the feeds in the open-source software R (R development core team 2006) based on 146 data points for LOAEL in fish and 111 data points for NOAEL from the same studies (Appendix A). The missing NOAEL levels were calculated by using the linear regression model implemented earlier by [148]. Afterwards, log CC5 were estimated based on the predicted NOAEL data sets using the Bayesian modelling as described by [148] and datapoints are available in Appendix A. The probability of CC5 estimate was given graphically by kernel density plot, which evaluated 100 equally spaced points that cover the range of the dataset. Also, boxplots were generated that show the distribution of the estimated CC5. Both, kernel density plots and boxplots were created in the software MATLAB (R2019b). The analyses have been performed for four different datasets separately: all fish species (*n* = 146), rainbow trout (*n* = 56), salmonids, (*n* = 67) and all fish species excluding rainbow trout (*n* = 90).

### 4.4. Meta-Analysis

To answer our research questions regarding the effects of DON on feed intake and growth performance in fish, a meta-analytical approach was performed in alignment with similar studies have done before in pigs and poultry [153,154,155,162,163]. Out of the 28 articles initially used in the systematic review, 12 experiments reported in 11 studies (rainbow trout; 7, salmon; 2, tilapia; 2, carp; 1 experiment included in the meta-analysis (Appendix A). We used the following selection criteria: (1) Only in vivo studies were selected. (2) In the selected studies DON concentration in the feed (µg/kg) was reported. (3) The studies aimed to investigate the single effects of DON in fish species. (4) Only studies that implemented satiation feeding were selected so that we could evaluate the effect of the toxin on feed intake capacity. (5) Average daily feed intake (g/fish/day) and average daily weight gain (g/fish/day) should be reported. A study could be included in our dataset only if both feed intake and growth data were available because apart from the effects of dietary DON on feed intake and growth, we also investigated the potential association between feed intake and growth. (6) Each study should include a control diet. (7) Exclusion of treatments that contained a feed additive against mycotoxin (e.g., mycotoxin binder) could moderate the effects of DON. (8) Inclusion of control and treatments diets with the presence of other toxins than DON. In these cases, naturally contaminated ingredients were used and the exclusion of other toxins was impossible. Consequently, the authors decided not to exclude these studies. (9) Inclusion of data on the response variables from different time points within a study, when there were reported.

Each line in our dataset was linked to a different treatment (control or DON treatments) coded as non-challenged (control) and challenged (DON treatments). The analysed independent (predictor) variable was the experimental concentration of DON (μg/kg) in the diets. The dependent (response) variables were average daily feed intake (feed intake) and average daily weight gain (growth). To reduce the variation effect among studies (fish species, experimental duration, fish size etc.), data were standardised by transforming feed intake and growth data relative to their controls; feed intake (% control) and growth (% control). For each study, additional information about the animal (fish species, number of fish per treatment, initial and final body weight) and the experimental conditions (duration of the exposure in days, feeding frequency etc.) was recorded in the database (Appendix A).

The meta-analysis was performed for all species collectively (63 data points), and rainbow trout separately (35 data points). Initially, in both databases the quality of the data was assessed graphically by scatter plots and the Spearman’s rho correlation test. Outliers were not removed as they might reflect pathological effects and the high variability in the experimental animals received a DON-contaminated diet. Afterwards, the effect of dietary DON on feed intake (% control) and growth (% control) was preliminarily assessed by regression analysis. For all cases, the outcomes showed that the DON impact on the dependent variables was explained better (>R^2^) by a quadratic relationship over a linear. However, a quadratic relationship could not physiologically explain the effect of DON on either feed intake or growth and, therefore, an exponential trendline was chosen. Finally, by using SAS software^®^ (version 9.4, SAS Institute, Cary, NC) and the NLIN (non-linear regression) procedure, exponential equations were estimated to describe the relation between predictor (x) and response variables (y):(y) = a ∗ eb(x)(1)
where (y) is the feed intake (% control) or growth (% control); (x) concentration of DON (mg/kg) in the feed; (a) the value for (y) when DON concentration is 0; and (b) the regression coefficient.

The pseudo-R2 to assess model fit was calculated as:Pseudo-R2 = 1 − (SSerror/SStotal(corrected))(2)
where SSerror is the error sums of squares and SStotal the total sum of squares.

For rainbow trout data, it was evaluated if the feed intake and growth response vary significantly between two types of DON; natural vs. pure by testing the difference in their regression coefficients. Finally, the relationship between feed intake (% control) and growth (% control) response in all fish species and rainbow trout was assessed by linear regression.

## Figures and Tables

**Figure 1 toxins-13-00403-f001:**
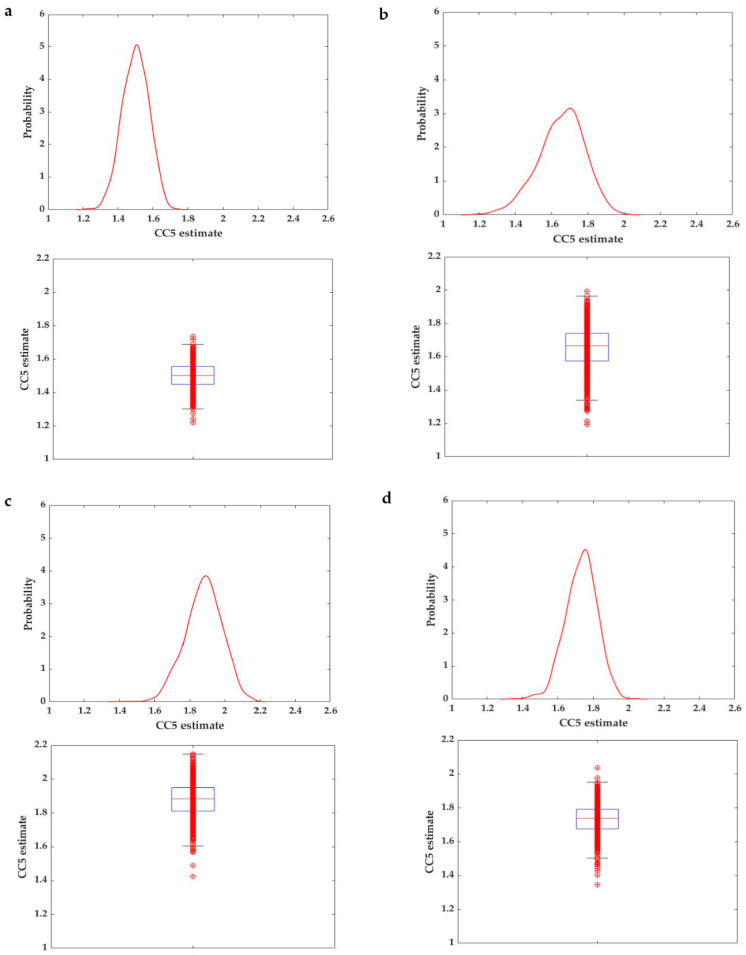
Kernel density plot with the probability of estimated log critical concentration 5% (CC5), and boxplots of log CC5 for DON exposure in (**a**) fish species, *n* = 146, (**b**) rainbow trout, *n* = 56, (**c**) salmonids, *n* = 67 and (**d**) all fish species excluding rainbow trout, *n* = 90.

**Figure 2 toxins-13-00403-f002:**
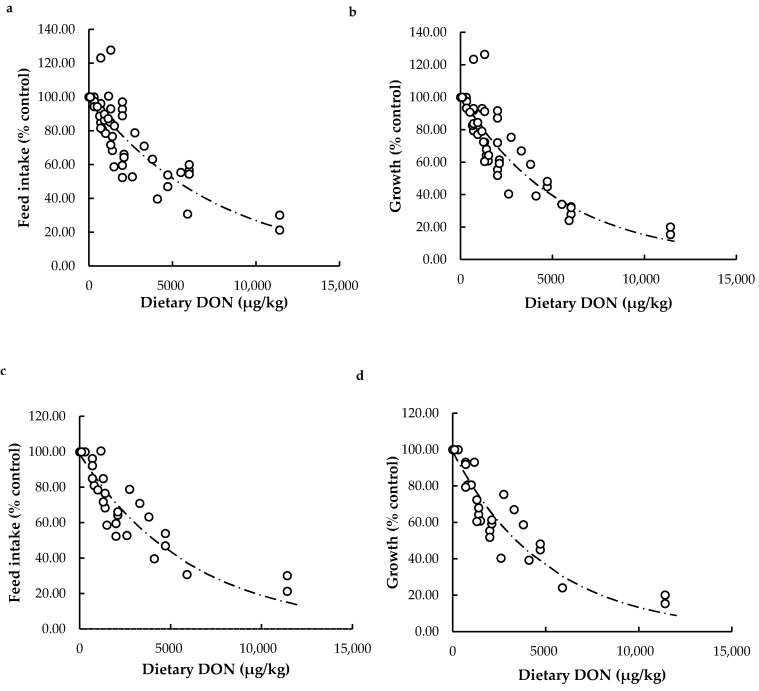
Effect of dietary DON concentration on feed intake (**a**) and growth (**b**) for all fish species in the dataset (*n* = 63) and for rainbow trout only (**c**,**d**; *n* = 35). Feed intake and growth values are expressed as percentage (%) of feed intake and growth seen in the control groups of the respective studies. The estimated relationships for all fish species were: (**a**) feed intake = 100.4 (±2.2) e^−0.132 (±0.013) × DON^, pseudo-R^2^ = 0.74; (**b**) growth = 99.0 (±2.6) e^−0.165 (± 0.016) × DON^, pseudo-R^2^ = 0.85. The estimated relationships for rainbow trout were: (**c**) feed intake= 101.1 (±2.3) e^−0.188 (±0.016) × DON^, pseudo-R^2^ = 0.81; (**d**) growth = 98.9 (±2.6) e^−0.200 (±0.018) × DON^, pseudo-R^2^ = 0.87. In all prediction equations above, DON concentration in the feed is expressed in mg/kg.

**Figure 3 toxins-13-00403-f003:**
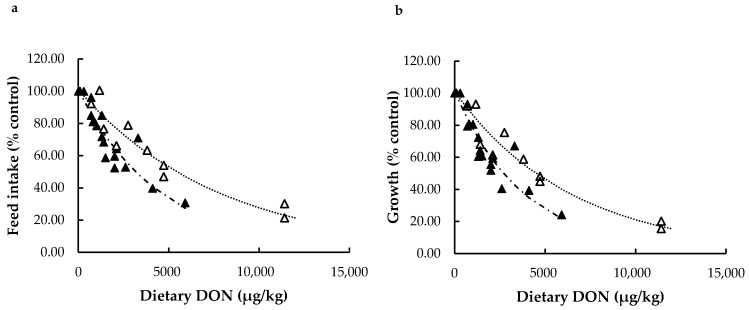
Effects of natural dietary DON (▲) and pure DON (△) on feed intake (**a**) and growth (**b**) in rainbow trout (*n* = 35). Feed intake and growth values were expressed as percentage (%) of the feed intake and growth of the control treatment in the respective studies. The estimated relationships were: (**a** ▲) feed intake = 100.9 (±1.5) e^−0.221^ (±0.016) × DON pseudo-R2 = 0.91; (**a** △) feed intake = 100.9 (±1.08) e^−0.129^ (±0.008) × DON pseudo-R^2^ = 0.96 and (**b** ▲) growth = 101.1 (±1.5) e^−0.260^ (±0.018) × DON pseudo-R^2^ = 0.92; (**b** △) growth= 100.9 (±1.1) e^−0.157^ (± 0.009) × DON pseudo-R^2^ = 0.97. In all prediction equations above, DON is the concentration in the feed expressed in mg/kg.

**Figure 4 toxins-13-00403-f004:**
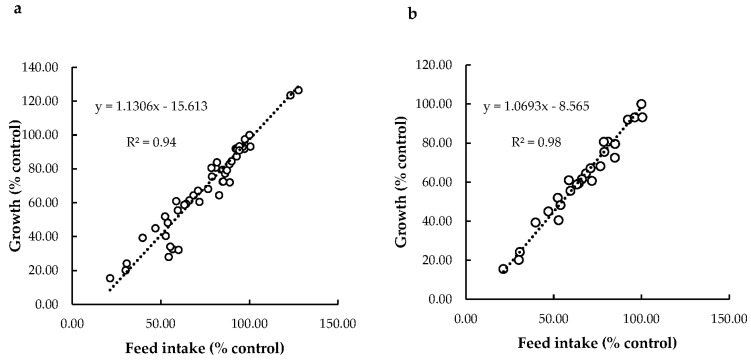
Relationship between feed intake (% control) and growth (% control) in: (**a**) all fish species in the dataset (*n* = 63); (**b**) rainbow trout (*n* = 35) fed diets with DON.

**Table 1 toxins-13-00403-t001:** Mycotoxins occurrence in wheat (*n* = 857), corn (*n* = 725) and soybean meal (*n* = 139) samples ^1^.

	Wheat			Corn			Soybean Meal		
Mycotoxin	Occurrence ^2^ (%)	Mean (µg/kg)	Maximum (µg/kg)	Occurrence ^2^ (%)	Mean (µg/kg)	Maximum (µg/kg)	Occurrence ^2^ (%)	Mean (µg/kg)	Maximum (µg/kg)
15-acetyl-deoxynivalenol (15-AcDON)	4	51	217	20	133	1667	1	13	13
3-acetyl-deoxynivalenol (3-AcDON)	7	28	101	14	46	406	.	.	.
Aflatoxin B_1_ (AFB_1_)	2	2	6	4	12	148	6	1	2
Aflatoxin B_2_ (AFB_2_)	3	9	51	4	19	92	2	4	5
Aflatoxin G_1_ (AFG_1_)	1	3	14	2	12	67	2	19	51
Aflatoxin G_2_ (AFG_2_)	3	3	14	8	7	60	1	2	2
Alternariol	12	30	247	3	20	110	9	27	109
Beauvericin (BEA)	1	5	14	5	56	552	4	11	27
Citreoviridin	0.1	1172	1172	0.1	33	33	.	.	.
Citrinin	1	9	17	0.3	10	18	5	84	224
Cyclopiazonic acid	1	19	44	2	16	73	2	19	30
Deoxynivalenol (DON)	41	470	8872	47	826	10,020	11	85	543
Diacetoxyscirpenol (DAS)	1	39	81	3	26	187	.	.	.
DON-3-Glucoside (DON3Glc)	7	137	1072	7	202	851	4	59	62
Ergocristin(in)e	0.4	95	189	.	.	.	.	.	.
Ergocryptin(in)e	0.5	10	25	.	.	.	.	.	.
Ergometrin(in)e	4	23	361	2	8	34	5	4	9
Ergosin(in)e	0.2	35	46	.	.	.	.	.	.
Ergotamin(in)e	3	119	1891	4	7	102	4	4	6
Fumonisin B_1_ (FB_1_)	27	561	9122	70	2234	49,347	26	371	1462
Fumonisin B_2_ (FB_2_)	14	59	590	54	262	7944	19	83	424
Fumonisin B_3_ (FB_3_)	4	67	417	41	189	3203	6	50	159
Fusarenon X (FX)	23	91	1267	10	96	604	12	65	196
Fusaric acid (FA)	5	54	337	67	266	4327	42	89	754
Gliotoxin	2	292	811	1	247	879	.	.	.
HT-2 toxin	4	44	456	9	190	2643	4	155	561
Lysergol	3	4	8	2	2	6	8	3	9
Methylergonovine	6	3	11	5	5	30	7	3	14
Moniliformin (MON)	1	14	24	10	171	1103	.	.	.
Mycophenolic acid	2	39	228	4	79	478	1	297	297
Neosolaniol (NEO)	6	18	79	8	48	589	8	26	158
Nivalenol (NIV)	1	275	453	4	661	1660	3	231	291
Ochratoxin A (OTA)	11	6	45	9	24	648	12	3	7
Ochratoxin B	5	3	9	6	4	53	6	3	6
Patulin	1	128	183	1	102	183	2	101	106
Penicillic acid	.	.	.	3	297	2156	.	.	.
Roquefortine C	5	3	26	10	4	71	10	2	5
Sterigmatocystin	6	4	21	8	2	5	12	2	4
T-2 toxin	7	46	551	14	81	757	23	49	348
Verruculogen	8	15	367	5	65	802	3	10	17
Wortmannin	4	39	474	2	124	508	1	25	28
Zearalanone	0.2	463	606	1	137	555	.	.	.
Zearalenone (ZEN)	5	64	738	16	165	1282	14	81	354

^1^ Mean and maximum values were calculated for the positive samples. ^2^ In case that a toxin was not detected in any of the samples (below the detection limits of analysis), the symbol “.” is used and represents 0% occurrence.

**Table 2 toxins-13-00403-t002:** Odds ratio (OR) of the association between deoxynivalenol (DON) and contamination with other toxins in corn.

“Toxin X”	Category	Frequency ^1^		% DON ^2^	Odds Ratio (OR)	95% CI	Wald *p*-Value ^3^
		*n*	%				
DON-3-Glucoside (DON3Glc)	present	53	7.3	98.1	69.6	9.6–505.9	***
	absent	672	92.7	42.7	Ref.		
15-acetyl-deoxynivalenol (15-AcDON)	present	142	19.6	96.5	51.7	20.8–128.2	***
	absent	583	80.41	34.7	Ref.		
Nivalenol (NIV)	present	30	4.1	96.7	36.0	4.9–265.9	***
	absent	695	95.9	44.6	Ref.		
3-acetyl-deoxynivalenol (3-AcDON)	present	98	13.5	91.8	17.1	8.1–35.8	***
	absent	627	86.5	39.7	Ref.		
Zearalenone (ZEN)	present	116	16.0	90.5	15.3	8.0–29.1	***
	absent	609	84.0	38.4	Ref.		
Sterigmatocystin	present	57	7.9	68.4	2.7	1.5–4.7	***
	absent	668	92.1	44.9	Ref.		
Roquefortine C	present	72	9.9	68.1	2.7	1.6–4.5	***
	absent	653	90.1	44.4	Ref.		
Alternariol	present	20	2.8	70.0	2.7	1.04–7.2	*
	absent	705	97.2	46.1	Ref.		
HT-2 Toxin	present	67	9.2	64.2	2.2	1.3–3.7	**
	absent	658	90.8	45.0	Ref.		
T-2 toxin	present	103	14.2	62.1	2.1	1.3–3.2	***
	absent	622	85.8	44.2	Ref.		
Fusarenon X (FX)	present	70	9.7	60.0	1.8	1.1–3.0	*
	absent	655	90.3	45.3	Ref.		
Neosolaniol (NEO)	present	57	7.9	59.7	1.8	1.01–3.1	*
	absent	668	92.1	45.7	Ref.		
Fumonisin B_1_ (FB_1_)	present	505	69.7	50.3	1.6	1.2–2.2	**
	absent	220	30.3	38.6	Ref.		
Fumonisin B_3_ (FB_3_)	present	296	40.8	53.0	1.5	1.1–2.1	**
	absent	429	59.2	42.4	Ref.		
Beauvericin (BEA)	present	36	5.0	25.0	0.36	0.17–0.78	**
	absent	689	95.0	47.9	Ref.		
Moniliformin (MON)	present	70	9.7	31.4	0.48	0.29–0.83	**
	absent	655	90.3	48.4	Ref.		
Aflatoxin B_2_ (AFB_2_)	present	29	4.0	31.0	0.49	0.22–1.1	#
	absent	696	96.0	47.4	Ref.		
Gliotoxin	present	5	0.7	80.0	4.6	0.51–41.3	NS
	absent	720	99.3	46.5	Ref.		
Zearalanone	present	8	1.1	75.0	3.5	0.69–17.3	NS
	absent	717	98.9	46.4	Ref.		
Lysergol	present	12	1.7	66.7	2.3	0.69–7.7	NS
	absent	713	98.3	46.4	Ref.		
Diacetoxyscirpenol (DAS)	present	25	3.5	60.0	1.7	0.77–3.9	NS
	absent	700	96.6	46.3	Ref.		
Methylergonovine	present	35	4.8	57.1	1.6	0.78–3.1	NS
	absent	690	95.2	46.2	Ref.		
Ochratoxin A (OTA)	present	62	8.6	56.5	1.5	0.91–2.6	NS
	absent	663	91.5	45.9	Ref.		
Ergotamin(in)e	present	30	4.1	56.7	1.5	0.73–3.2	NS
	absent	695	95.9	46.3	Ref.		
Verruculogen	present	37	5.1	54.1	1.4	0.70–2.6	NS
	absent	688	94.9	46.4	Ref.		
Aflatoxin G_1_ (AFG_1_)	present	13	1.8	53.9	1.3	0.45–4.0	NS
	absent	712	98.2	46.6	Ref.		
Aflatoxin B_1_ (AFB_1_)	present	27	3.7	51.9	1.2	0.57–2.7	NS
	absent	698	96.3	46.6	Ref.		
Aflatoxin G_2_ (AFG_2_)	present	61	8.4	50.8	1.2	0.71–2.0	NS
	absent	664	91.6	46.4	Ref.		
Fusaric acid (FA)	present	485	66.9	48.0	1.2	0.86–1.6	NS
	absent	240	33.1	44.2	Ref.		
Ochratoxin B	present	44	6.1	50.0	1.1	0.62–2.1	NS
	absent	681	93.9	46.6	Ref.		
Ergometrin(in)e	present	12	1.7	50.0	1.1	0.37–3.6	NS
	absent	713	98.3	46.7	Ref.		
Citreoviridin	present	1	0.1	100.0	1.1	0.06-∞	NS ^4^
	absent	724	99.9	46.8	Ref.		
Mycophenolic acid	present	31	4.3	48.4	1.1	0.52–2.2	NS
	absent	694	95.7	46.7	Ref.		
Fumonisin B_2_ (FB_2_)	present	394	54.3	47.0	1.0	0.76–1.4	NS
	absent	331	45.7	46.5	Ref.		
Wortmannin	present	17	2.3	47.1	1.0	0.39–2.7	NS
	absent	708	97.7	46.8	Ref.		
Patulin	present	5	0.7	20.0	0.28	0.03–2.5	NS
	absent	720	99.3	46.7	Ref.		
Citrinin	present	2	0.3	0.0	0.47	0.0–4.0	NS ^4^
	absent	723	99.7	46.9	Ref.		
Cyclopiazonic acid	present	16	2.2	43.8	0.88	0.33–2.4	NS
	absent	709	97.8	46.8	Ref.		
Penicillic acid	present	24	3.3	45.8	0.96	0.43–2.2	NS
	absent	701	96.7	46.8	Ref.		

^1^ n refers to the number of samples where “Toxin X” is present or absent, and % is the percentage of frequency relative to the total number of corn samples. ^2^ Percentage of cases that DON exists with ”Toxin X” present, and cases with “Toxin X” absent. ^3^ Wald Chi-Square Test: Not significant (NS): *p* ≥ 0.1, # *p* <0.1, * *p* <0.05, ** *p* <0.01, *** *p* < 0.001. ^4^ Estimated with Exact Logistic Regression.

**Table 3 toxins-13-00403-t003:** Mycotoxins occurrence in aquafeed samples ^1^ (*n* = 44).

Mycotoxin	Occurrence (%)	Mean (µg/kg)	Maximum (µg/kg)
15-acetyl-deoxynivalenol (15-AcDON)	5	82	127
Aflatoxin B_1_ (AFB_1_)	5	2	4
Aflatoxin G_2_ (AFG_2_)	2	6	6
Alternariol	14	21	51
Deoxynivalenol (DON)	48	136	469
Diacetoxyscirpenol (DAS)	2	9	9
DON-3-Glucoside (DON3Glc)	18	98	155
Ergometrin(in)e	7	4	5
Ergotamin(in)e	20	38	125
Fumonisin B_1_ (FB_1_)	36	628	4923
Fumonisin B_2_ (FB_2_)	27	120	778
Fumonisin B_3_ (FB_3_)	11	86	223
Fusarenon X (FX)	2	28	28
Fusaric acid (FA)	55	41	265
Gliotoxin	2	92	92
HT-2 toxin	2	43	43
Lysergol	9	10	23
Ochratoxin A (OTA)	2	3	3
Penicillic acid	11	41	58
Sterigmatocystin	2	1	1
T-2 toxin	2	46	46
Verruculogen	9	560	636
Wortmannin	2	20	20
Zearalenone (ZEN)	2	348	348

^1^ Mean and maximum values were calculated for the positive samples.

**Table 4 toxins-13-00403-t004:** Descriptive data ^1^ of control and DON-challenged fish in the two meta-analyses combining information on all fish species (*n* = 63), or only rainbow trout (*n* = 35).

**All Fish Species**	**Control**	**Challenged**
Initial body weight (g)	30.90 ± 6.74	27.85 ± 3.96
DON dose (µg/kg)	70.61 ± 22.03	2575.04 ± 383.32
Feed intake (g/fish)	1.30 ± 0.17	0.98 ± 0.08
Growth (g/day)	1.27 ± 0.19	0.86 ± 0.08
**Rainbow Trout**	**Control**	**Challenged**
Initial body weight (g)	29.25 ± 9.95	29.41 ± 5.70
DON dose (µg/kg)	97.40 ± 36.19	2994.52 ± 581.40
Feed intake (g/fish)	1.51 ± 0.30	1.05 ± 0.14
Growth (g/day)	1.54 ± 0.33	1.00 ± 0.14

^1^ Mean ± standard error.

## Data Availability

The corresponding author can be contacted if access to the data is desired.

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
