# Peer review of "The Occurrence of Mycotoxins in Raw Materials and Fish Feeds in Europe and the Potential Effects of Deoxynivalenol (DON) on the Health and Growth of Farmed Fish Species—A Review"

_toxins, 2021, doi:10.3390/toxins13060403_

Round 1
Reviewer 1 Report
The manuscript entitled „The occurence of mycotoxins in raw materials and fish feeds in Europe and the potential effect of deoxynivalenol (DON) on the health and growth of farmed fish species” is an interesting combination of a scientific article and a review article.
In its first part it deals with occurrence and co-occurrence of mycotoxins in a large number of feed ingrediens (wheat, corn, soybean meal) and fish feeds used in Europe from the period between 2012-2019. Its second part is a literature review based on a remarkable amount of publications on the effect of DON on fish species, focusing on the most important species of aquaculture as carp, Nile tilapia, salmon, rainbow trout. The third part is a meta-analytical approach about the effect of DON on feed intake and growth of fish species (based on the data given in 63 well chosen literature) and highlighted on rainbow trout (based on 35 literature).
The text corpus of this manuscript consists of 29 pages and in the References a total of 173 relevant publications are listed.
Authors also mention 7 supplementary tables, but I could not review them, as they were not included in this manuscript.
The applied methodology and statistics for the meta-analytical calculations are adequate.
For those, who are interested in this field this excellent work will be a fundamental and integral summary of the data from relevant publications.
I congratulate to the authors for this huge work and after minor, but in some situation important changes, I advice this manuscript to be published in Toxins.
The most important mistake in this manuscript, which should be corrected is written at L258-259. This sentence is a serious problem, as corn is not a protein source, since its protein content is lower than 10% (not 60%!!!), it is a good source of energy (starch).
Some minor errors need to be corrected.
L117 - oC, rel H
Table 1.: max. occurence mean max. - bold
If the analysed wheat, corn samples came from the period between 2012-2019, and in an article published in 2021, a more up-to-date production values would be better (not from 2009) [70].
L318 – FumB3 - correct: FB3
L334 – correct: up to 30-45%
L361: correct: T-2
Table 1, Table 3 – the grouping criteria for the mycotoxins is not clear for me.
Alphabetic order or other aspect, eg. producing species (if so, AF and sterigmatocystin should be closer to each other)
L418: aflatoxin
L435, L838 . .
Table 2. 95% CI for DON3Glc and NIV is extremly wide. Are these high values correct?
Table 2, Table 3: The font size in the table should be similar to Table 1
L552: correct: >5000
L570: the ADC abbreviation is not necessary
L570: crude protein was mentioned earlier, so this CP abbreviation is not necessary here
The abbreviated name of the genes should be italic. (L627,L631, L633)
L661: correct: unstimulated
L669: correct: Reduced ROS … , as the abbreviation was mentioned earlier.
L676: correct: vs. (italic) also for L884, L1171
[148] – this is a study of combination of DON and ZEA
L782 - Correct: metabolisation
L809-L810 should be deleted.
L810 – Our systemati – where does this belong?
Fig. 1. - use the same scale of x and y-axis for all probability graph
Fig. 2. - the DON concentrations are missing (x-axis) – L871-874 can not be verified
c and d graphs for rainbow trout are missing (L857)
L827 - 2.2.3.1 – format this paragraph
Fig. 4. the data of statistical analysis should be centered in the diagrams, R2 values should be written at least 2 digits (if they are discussed at L938-939)
L960: fish species – correct: all fish species in the dataset
L965: and we thus we - correct: and thus we
Please, rephrase this sentence (L978-L980).
L482, L521, L714, L782, L822, L828, L993, L1092, L1126 - in vivo - italic
L712, L781 - in vitro - italic
L1110 - LOEL - correct: LOAEL
L1124 - the sum of the studies mentioned is not 11
L1166 - a , b -- correct: (a) (b)
The reference list should be checked, and corrected based on the guide for authors.
16 – add more informations!
19, 31, 32, 39, 43, 51, 55, 60, 69, etc. – correct the capital letters
40 , 50, 54, 56, etc. – correct the journal names
Food additives and contaminants vs. Food Additives & Contaminants
Poultry science vs. Poultry Science; etc.
Reviewer 2 Report
Mycotoxins are a growing global problem, causing enormous losses in agricultural crop yields and economy. Thus, the discussions about the extention of this critical issue are necessary and of great importance. The presented manuscript provides an interesting and comprehensive overview of the current data of mycotoxin occurrence. And moreover, it focuses on the mycotoxin presence in aquafeed which is rarely brought up in the discussions regarding mycotoxin food/feed contamination yet it is equally important. For this reason, manuscript is well-written and ready for the publication. There are a few issues that need to be addressed first:
- Line 69 - 185: Chapters 1.1-1.3 are too long and contain a lot of information that was published in many other similar publications and are considered as 'common facts' about mycotoxins. As this review is already very long, I would suggest you to shorten these chapters and merge them into only one titled 'the problem of mycotoxin contamination' or anything alike.
- Line 435: Table 1 - remove the second dot at the end of the table caption.
- Line 808 - end of the manuscript: correct the text form as the captions are not next to the corresponding figures and so on.
